# Transferrin receptor 1-mediated iron uptake regulates bone mass in mice via osteoclast mitochondria and cytoskeleton

Bhaba K Das[1†], Lei Wang[2,3†], Toshifumi Fujiwara[3,4†], Jian Zhou[5], Nukhet Aykin-Burns[6], Kimberly J Krager[6], Renny Lan[7], Samuel G Mackintosh[8], Ricky Edmondson[8], Michael L Jennings[9], Xiaofang Wang[10], Jian Q Feng[10], Tomasa Barrientos[11], Jyoti Gogoi[1], Aarthi Kannan[1,12], Ling Gao[1,12], Weirong Xing[13], Subburaman Mohan[13]*, Haibo Zhao[1,3,9]*

[1]Southern California Institute for Research and Education, Long Beach, United States; [2]Department of Orthopedics, The Third People's Hospital of Hefei, Third Clinical College, Anhui Medical University, Hefei, China; [3]Center for Osteoporosis and Metabolic Bone Diseases, Division of Endocrinology, Department of Internal Medicine, University of Arkansas for Medical Sciences, Little Rock, United States; [4]Department of Orthopedic Surgery, Kyushu University Hospital, Fukuoka, Japan; [5]Department of Orthopedics, First Affiliated Hospital, Anhui Medical University, Hefei, China; [6]Division of Radiation Health, Department of Pharmaceutical Sciences, University of Arkansas for Medical Sciences, Little Rock, United States; [7]Department of Pediatrics, University of Arkansas for Medical Sciences, Little Rock, United States; [8]Department of Biochemistry and Molecular Biology, University of Arkansas for Medical Sciences, Little Rock, United States; [9]Department of Physiology and Cell Biology, University of Arkansas for Medical Sciences, Little Rock, United States; [10]Department of Biomedical Sciences, Texas A&M University, Dallas, United States; [11]Department of Orthopedics, Duke University, Durham, United States; [12]Division of Dermatology, Department of medicine, Long Beach VA Healthcare System, Long Beach, United States; [13]Musculoskeletal Disease Center, VA Loma Linda Healthcare System, Loma Linda, United States

*For correspondence:
Subburaman.Mohan@va.gov (SM);
Haibo.zhao@va.gov (HZ)

[†]These authors contributed equally to this work

**Abstract** Increased intracellular iron spurs mitochondrial biogenesis and respiration to satisfy high-energy demand during osteoclast differentiation and bone-resorbing activities. Transferrin receptor 1 (Tfr1) mediates cellular iron uptake through endocytosis of iron-loaded transferrin, and its expression increases during osteoclast differentiation. Nonetheless, the precise functions of Tfr1 and Tfr1-mediated iron uptake in osteoclast biology and skeletal homeostasis remain incompletely understood. To investigate the role of Tfr1 in osteoclast lineage cells in vivo and in vitro, we crossed *Tfrc* (encoding Tfr1)-floxed mice with *Lyz2 (LysM)*-Cre and *Cathepsin K* (*Ctsk*)-Cre mice to generate *Tfrc* conditional knockout mice in myeloid osteoclast precursors (Tfr1[ΔLysM]) or differentiated osteoclasts (Tfr1[ΔCtsk]), respectively. Skeletal phenotyping by μCT and histology unveiled a significant increase in trabecular bone mass with normal osteoclast number in long bones of 10-week-old young and 6-month-old adult female but not male Tfr1[ΔLysM] mice. Although high trabecular bone volume in long bones was observed in both male and female Tfr1[ΔCtsk] mice, this phenotype was more pronounced in female knockout mice. Consistent with this gender-dependent phenomena, estrogen deficiency induced by ovariectomy decreased trabecular bone mass in Tfr1[ΔLysM] mice. Mechanistically, disruption of Tfr1 expression attenuated mitochondrial metabolism and cytoskeletal organization in mature osteoclasts in vitro by attenuating mitochondrial respiration and activation

of the Src-Rac1-WAVE regulatory complex axis, respectively, leading to decreased bone resorption with little impact on osteoclast differentiation. These results indicate that Tfr1-mediated iron uptake is specifically required for osteoclast function and is indispensable for bone remodeling in a gender-dependent manner.

## Editor's evaluation

The current work addresses an important unknown in bone homeostasis, aiming to understand the mechanism of iron-mediated effects on bone, and presents findings of novel significance that are of interest to basic iron biologists, bone biologists, experts in mitochondrial respiration, and endocrinologists. The authors specifically focus on the role of iron deficiency via TFR1 loss in osteoclastogenesis and osteoclast function, demonstrating that cell-selective loss of TFR1 in mature osteoclasts (using Ctsk-Cre mice) leads to decreased osteoclast function and increased bone volume in long bones, especially in female mice, without influencing osteoclastogenesis. In vitro studies confirm an actin cytoskeletal defect and defect in bone resorption in TFR1 null osteoclasts, both partially rescued by heme but not NTBI. Together, these findings contribute to the knowledge in the field, supporting a hypothesis that iron delivered via TF-TFR1 (or heme iron) is essential for osteoclast function.

## Introduction

Bone mass and structure in adults are maintained by constant bone remodeling with balanced bone formation by osteoblasts and bone resorption by osteoclasts (*Zaidi, 2007*; *Siddiqui and Partridge, 2016*). Under pathological conditions, however, excessive bone resorption caused by either increased number or exaggerated activities of osteoclasts leads to bone loss which is a hallmark of many metabolic bone diseases such as osteoporosis, rheumatoid arthritis, Paget's disease of bone, periodontal disease, and tumor bone metastasis (*Mbalaviele et al., 2017*; *Novack and Teitelbaum, 2008*). Osteoclasts are multinucleated cells formed by fusion of mononuclear precursors that are differentiated from the monocyte/macrophage lineage of hematopoietic cells. This process is triggered by two indispensable cytokines M-CSF (macrophage colony-stimulating factor) and RANKL (receptor activator of NF-κB ligand) (*Walker, 1975*; *Xiao et al., 2017*; *Boyle et al., 2003*; *Nakashima et al., 2012*). Upon attachment to bone, osteoclasts organize their actin cytoskeleton to form actin-rings that seal the resorptive microenvironment (*Väänänen et al., 2000*; *Teitelbaum, 2011*). The dissolution of bone minerals and digestion of organic bone matrix mainly type I collagen are executed by hydrochloric acid and acidic hydrolase cathepsin K, respectively (*Zhao, 2012*; *Fujiwara et al., 2016*). Both osteoclast formation and bone-resorbing activity of mature osteoclasts demand high energy. However, the pathways and molecular mechanisms regulating osteoclast energy metabolism in osteoclasts remain largely unknown.

Both glycolysis and mitochondrial oxidative phosphorylation (OXPHOS) increase during osteoclast differentiation (*Kubatzky et al., 2018*; *Arnett and Orriss, 2018*; *Kim et al., 2007*; *Indo et al., 2013*; *Li et al., 2020a*). Osteoclasts contain numerous mitochondria (*Ch'uan, 1931*; *Holtrop and King, 1977*). The mitochondrial respiratory complex I and the key mitochondria transcriptional regulator PGC-1β (peroxisome proliferator-activated receptor γ coactivator 1β) are crucial for osteoclast differentiation (*Jin et al., 2014*; *Ishii et al., 2009*). Mitochondrial ROS (reactive oxygen species) stimulates osteoclast differentiation (*Srinivasan et al., 2010*; *Kim et al., 2013*; *Kim et al., 2017*). On the contrary, it has been reported that decreased mitochondrial biogenesis and activity by PGC-1β deletion in osteoclast lineage cells disrupt osteoclast cytoskeletal organization and function but not osteoclastogenesis (*Zhang et al., 2018*). Loss of G-protein Gα13 in osteoclast progenitors promotes mitochondrial respiration and cytoskeleton organization but has little effects on osteoclastogenesis in cultures and in mice (*Nakano et al., 2019*). Heretofore, the mechanisms by which energy metabolism regulates osteoclast differentiation and function remain unclear.

Iron is a nutritional element that plays a fundamental role in mitochondrial metabolism and the biosynthesis of heme and Fe-S clusters which are critical components of the mitochondrial respiratory complexes (*Figure 1—figure supplement 1A* and *Xu et al., 2013*). Both systemic and cellular iron homeostasis play a pivotal role in bone remodeling (*Balogh et al., 2018*). Osteoporosis and

pathological bone fractures are commonly associated with the hereditary iron-overload disease hemochromatosis and the acquired iron-overload conditions in treatment of thalassemia and sickle cell disease (*Almeida and Roberts, 2005*; *Fung et al., 2008*; *Dede et al., 2016*; *Jandl et al., 2020*). Moreover, increased bone resorption and/or decreased bone formation have been observed in genetic mouse models of iron overload diseases and in mice fed with excessive iron (*Tsay et al., 2010*; *Guggenbuhl et al., 2011*; *Xiao et al., 2015*). In contrast to iron overload, much less is known about the influence of iron deficiency on bone cells and bone homeostasis. Severe iron deficiency impairs both bone resorption and bone formation and causes osteopenia in rats (*Katsumata et al., 2009*; *Diaz-Castro et al., 2012*), whereas iron chelation inhibits osteoclastogenesis and bone resorption in vitro and in vivo and attenuates estrogen deficiency induced bone loss in mice (*Ishii et al., 2009*; *Guo et al., 2015*).

There are two transferrin (Tf) receptors (Tfrs) in mammalian cells (*Kawabata, 2019*). Transferrin receptor 1 (Tfr1), encoded by *Tfrc* gene, is ubiquitously expressed with high affinity to $Fe^{3+}$-loaded holo-Tf. Germline deletion of *Tfrc* in mice leads to early embryonic lethality due to severe defects in erythroid and neuronal development (*Levy et al., 1999*). A missense mutation in *TFRC* causes combined immunodeficiency in human (*Jabara et al., 2016*). The tissue-specific deletion of *Tfrc* gene in neurons, skeletal muscles, cardiomyocytes, hematopoietic stem cells, and adipocytes in mice disrupts cellular iron homeostasis and causes severe metabolic defects in these tissues (*Matak et al., 2016*; *Barrientos et al., 2015*; *Xu et al., 2015*; *Wang et al., 2020*; *Li et al., 2020b*). In contrast to Tfr1, Tfr2 has lower affinity to holo-Tf and is predominantly expressed in liver and erythroid precursor cells (*Kawabata, 2019*). Mutations of *TFR2* in human and germline or hepatocyte-specific deletion of *Tfr2* in mice cause type 3 hemochromatosis (*Camaschella et al., 2000*; *Fleming et al., 2002*). More recently, Tfr2 has been reported to regulate osteoblastic bone formation and bone mass in mice, independent of its function in iron homeostasis (*Rauner et al., 2019*). Nevertheless, the cell autonomous functions of Tfr1 and Tfr2 in osteoclast lineage cells remain unknown.

To elucidate the role of Tfr1 and Tfr1-mediated iron uptake in osteoclast lineage cells, we generated *Tfrc* conditional knockout mice in myeloid osteoclast precursors and differentiated osteoclasts by crossing *Tfrc*-flox mice with *Lyz2*-Cre and *Ctsk*-Cre mice, respectively, and used them for comprehensive skeletal phenotyping in vivo and mechanistic studies in vitro.

## Results

### Tfr1-mediated iron uptake is the major route for iron acquiring in murine osteoclasts

Mammalian cells acquire iron through the uptake of Tf, heme, ferritins, and by non-transferrin bound iron (NTBI) (*Figure 1—figure supplement 1B*; *Pantopoulos et al., 2012*). In Tf-dependent pathway, holo-Tf binds to Tfrs on cell surface, and the complex is then internalized via endocytosis. $Fe^{3+}$ is released from Tf in endosomes and is reduced to ferrous iron ($Fe^{2+}$) by the STEAP (six-transmembrane epithelial antigen of the prostate) family of metaloreductases before being transported to the cytoplasm via DMT1 (divalent metal transporter 1) (*Gammella et al., 2017*). Cellular iron that is not utilized is either stored in ferritins or is exported via ferroportin (FPN) (*Donovan et al., 2006*; *Hentze et al., 2010*). In this study, we first set out to determine which of the four cellular iron uptake pathways operate in osteoclasts. For this purpose, we quantified the mRNA level of genes encoding the major transporters along each iron acquiring pathway (*Figure 1—figure supplement 1B*) by real-time PCR in three different stages of murine osteoclast lineage cells: bone marrow monocytes (BMM), mononuclear osteoclast precursors, and multinucleated mature osteoclasts. By this assay, we observed that the expression of genes participated in Tf/Tfr-mediated iron uptake: *Tfrc*, *Tfr2*, and *Slc11a2* (encoding Dmt1) were upregulated in mature osteoclasts (*Figure 1A*). We have previously reported that Steap4 is highly expressed in mature osteoclasts and plays an important role in cellular iron homeostasis in osteoclasts in vitro (*Zhou et al., 2013*). Together, all key mediators of Tf-dependent iron uptake pathway are upregulated during osteoclast differentiation. In contrast, the mRNA expression of *Slc39a14* (encoding Zip14, a transporter mediating cellular iron entrance via NTBI) was decreased during osteoclast differentiation. The expression of *Slc46a1*, which encodes the heme transporter Hcp1, was slightly increased in mature osteoclasts. The mRNA level of *Flvcr2*, which encodes another heme transporter, was low and did not change during osteoclast differentiation (*Figure 1A*). The

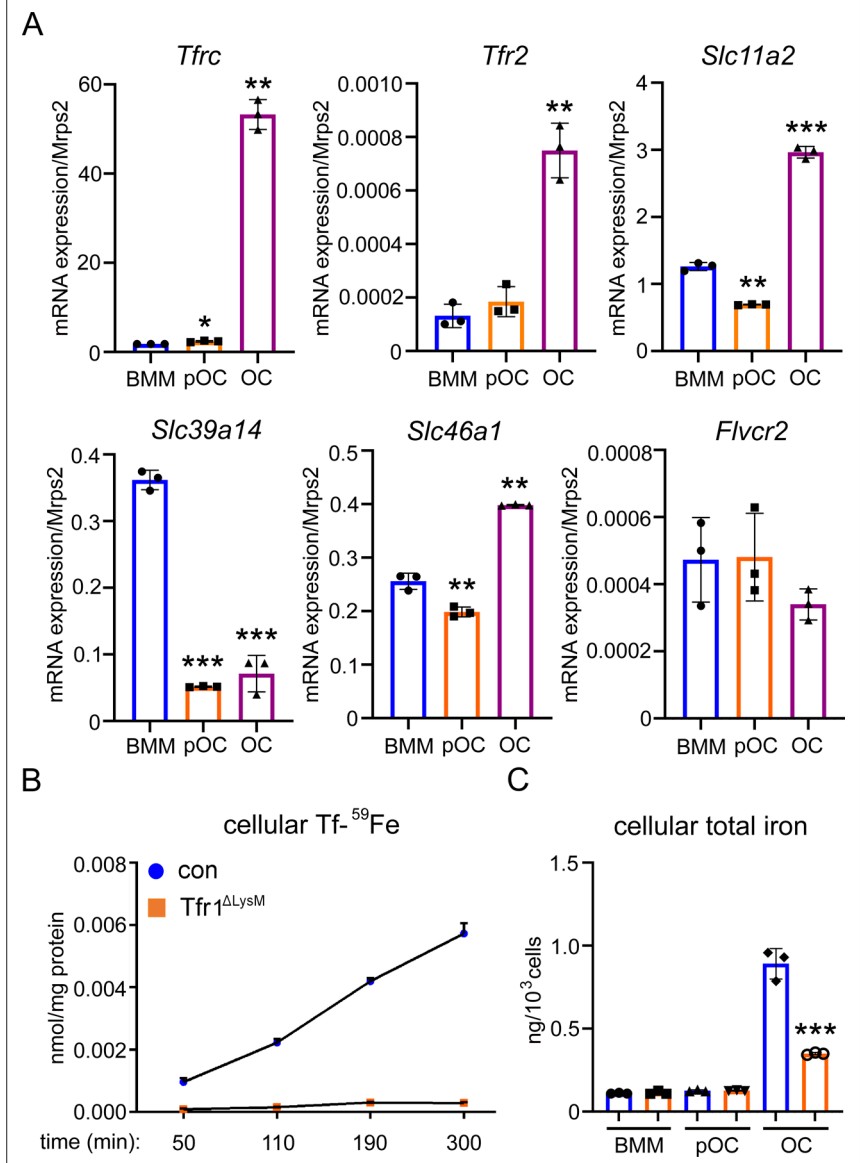

**Figure 1.** Tfr1 is a major transferrin transporter and regulates cellular iron homeostasis in osteoclasts. (**A**) Detection of mRNA expression of genes involved in iron uptake and export pathways in bone marrow monocytes (BMM), mononuclear pre-osteoclasts (pOC), and mature osteoclasts (OC) by real-time quantitative PCR. *Tfrc* encodes Tfr1; *Slc11a1* encodes Dmt1 (divalent metal ion transporter 1); *Slc39a14* encodes Zip14; *Slc46a1* encodes heme transporter Hcp1; *Flvcr2* encodes Flvcr2 heme transporter 2. (**B**) Measurement of $^{59}$Fe-labeled transferrin (Tf-$^{59}$Fe) uptake in control (con) and Tfr1 myeloid conditional knockout (Tfr1$^{\Delta LysM}$) osteoclasts by a gamma counter. (**C**) Measurement of intracellular total iron by a colorimetric iron assay kit in control and Tfr1-deficient osteoclast lineage cells. The data are presented as mean ± SD. n=3. * $p<0.05$, ** $p<0.01$, and *** $p<0.001$ vs BMM (**A**) and vs control OC (**C**) by one-way ANOVA and Student's t-test.

The online version of this article includes the following source data and figure supplement(s) for figure 1:

**Source data 1.** Tfr1 is a major transferrin transporter and regulates cellular iron homeostasis in osteoclasts.

**Figure supplement 1.** Graphic diagrams of iron in energy metabolism in mammalian cells (**A**) and cellular iron-uptake pathways (**B**).

mRNAs of *Timd2* and *Scara5*, which encode two plasma membrane transporters of ferritin, were undetectable in osteoclast lineage cells (data not shown).

Although both *Tfrc* and *Tfr2* mRNA expression increase during osteoclastogenesis, the level of *Tfr2* mRNA was much lower than that of *Tfrc* (*Figure 1A*). Moreover, deletion of Tfr1 completely

eliminated the uptake of Tf-$^{59}$Fe in mature osteoclasts, indicating that Tfr1 is the predominant receptor for Tf-uptake in osteoclasts (*Figure 1B*). Despite the previous finding that Tfr2-deletion in monocytes has no effects on iron homeostasis (*Rishi et al., 2016*), the role of Tfr2 in osteoclast iron uptake and function in vitro and in vivo needs to be further investigated in future. There was a dramatic increase in cellular iron concentration in mature osteoclasts compared to monocytes and pre-osteoclasts. Loss of Tfr1 in osteoclasts led to a >50% decrease in total intracellular iron content (*Figure 1C*), suggesting that Tf-dependent iron uptake contributes remarkably to iron acquirement in mature osteoclasts. We and others have previously found that the mRNA expression of *Slc40a1* (encoding Fpn, the only iron exporter identified so far in mammalian cells) decreases during osteoclast differentiation (*Gu et al., 2015*; *Wang et al., 2018*) and loss of Fpn in myeloid osteoclast precursors increases cellular iron and stimulates osteoclastogenesis in vitro and in vivo (*Wang et al., 2018*). Therefore, by transcriptional upregulation of Tf-dependent iron uptake pathway and concomitant downregulation of iron exporter Fpn, osteoclasts augment their cellular iron level to meet high energy demand during osteoclast differentiation and bone resorption. Nevertheless, the role of Tfr1 and Tf-dependent iron uptake in osteoclast biology and bone remodeling has not been elucidated using genetically modified mouse models.

## Loss of Tfr1 in myeloid osteoclast precursors increases trabecular bone mass of long bones prominently in adult female mice

The *Tfrc* germline deletion mice are embryonically lethal (*Levy et al., 1999*). To elucidate the role of Tfr1 in osteoclast differentiation and bone resorption during post-natal bone modeling and remodeling, we generated *Tfrc* myeloid conditional knockout mice on C57BL6/J background by crossing *Tfrc*-flox mice, in which the exons 3–6 of murine *Tfrc* gene were flanked by two loxP sites (*Chen et al., 2015*), with *Lyz2 (LysM)*-Cre mice. Our pilot study demonstrated that the skeletal phenotypes of three lines of control mice: *Tfrc*$^{flox/flox}$;*Lyz2*$^{+/+}$, Tfrc$^{+/+}$;*Lyz2*$^{Cre/Cre}$, and *Tfrc*$^{+/+}$;*Lyz2*$^{+/+}$ male and female mice were indistinguishable (*Figure 2—figure supplement 1*). We also noticed that single allele of *Lyz2*-Cre (*Tfrc*$^{flox/flox}$;*Lyz2*$^{Cre/+}$) only led to partial deletion of Tfr1 in osteoclast lineage cells, whereas two copies of *Lyz2*-Cre (*Tfrc*$^{flox/flox}$;*Lyz2*$^{Cre/Cre}$) completely eliminated Tfr1 in osteoclasts detected by western blotting (see Figure 4C). Moreover, trabecular bone mass and structure in *Tfrc*$^{flox/flox}$;*Lyz2*$^{Cre/+}$ mice evaluated by μCT were similar to their gender-matched littermate control mice (*Figure 2—figure supplement 2*). Therefore, we used *Tfrc*$^{flox/flox}$;+/+ and *Tfrc*$^{flox/flox}$;*Lyz2*$^{Cre/Cre}$ mice as control and Tfr1 myeloid conditional knockout (Tfr1$^{ΔlysM}$) mice, respectively, for further in vivo and in vitro studies.

Both male and female Tfr1$^{ΔlysM}$ mice were born at expected Mendelian ratio and developed normally with similar body size and weight compared to their littermate controls at 3 weeks, 10 weeks, and 6 months ages (data not shown). The serum levels of total iron and iron-regulating hormone hepcidin as well as the number of red blood cells in circulation of male and female Tfr1$^{ΔlysM}$ mice were similar compared to their respective controls (*Figure 2—figure supplement 3*), indicating that loss of Tfr1 in myeloid lineage cells has no effects on systemic iron homeostasis and erythropoiesis.

The μCT analysis of distal femurs isolated from 3-week-old control, and Tfr1$^{ΔlysM}$ mice revealed that loss of Tfr1 in osteoclast myeloid precursor cells at pre-pubertal age led to a mild increase in trabecular bone mass (bone volume/tissue volume [BV/TV]), trabecular number (Tb.N), and a decrease in trabecular spacing (Tb.Sp) in male Tfr1$^{ΔlysM}$ mice compared to gender matched control mice. Only a slight increase in trabecular thickness (Tb.Th) was observed in pre-pubertal female Tfr1$^{ΔlysM}$ mice compared to control mice (*Figure 2* and *Figure 2—figure supplement 4*). At 10-week post-pubertal age, however, trabecular BV/TV (>twofold), Tb.N, and Tb.Th were increased while Tb.Sp was deceased in the distal femurs of female Tfr1$^{ΔlysM}$ mice compared to controls. The increased trabecular bone mass phenotype in female Tfr1$^{ΔlysM}$ mice was retained at 6 months of age (*Figure 2A-E*). There was no difference in any of the above trabecular bone parameters between control and Tfr1$^{ΔlysM}$ male mice at either 10 weeks or 6 months age (*Figure 2—figure supplement 4A–4E*). The trabecular bone mass was significantly higher in male control mice after puberty. Since increased trabecular bone volume only occurred in Tfr1-deficient female mice, the trabecular bone mass of Tfr1$^{ΔlysM}$ female mice at 10 week and 6 month of ages was similar to age-matched control as well as Tfr1$^{ΔlysM}$ male mice. No changes in the cortical bone thickness (Cort.Th) were detected by μCT in the mid-shaft of femurs of both male and female Tfr1$^{ΔlysM}$ mice at all three ages compared to their respective controls (*Figure 2F* and *Figure 2—figure supplement 4F*). The μCT analysis of tibias showed similar trabecular bone mass

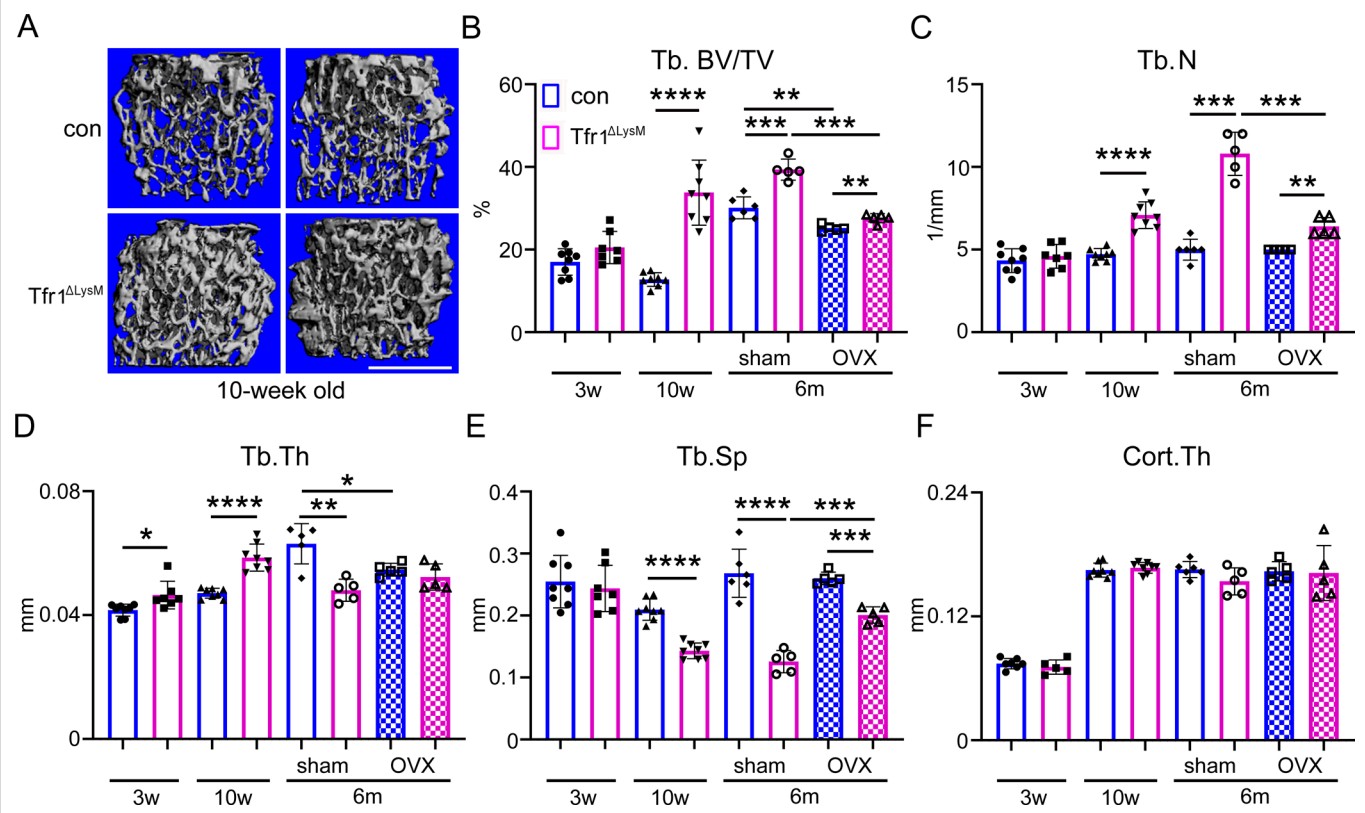

**Figure 2.** *Tfr1* myeloid conditional knockout female mice develop normally and display increased trabecular bone mass in femurs at 10-week-old and 6-month-old ages. (**A**) Representative μCT images of distal femurs of 10-week-old female control (con) and Tfr1^ΔLysM mice. scale bar = 1 mm. (**B– F**) μCT analysis of trabecular and cortical bone mass and structure of distal femurs isolated from three different ages of female con and Tfr1^ΔLysM mice. Tb: trabecular bone; BV/TV: bone volume/tissue volume; Tb.N: trabecular number; Tb.Th: trabecular thickness; Tb.Sp: trabecular spacing; Cort.Th: cortical bone thickness. The data are presented as mean ± SD. n=5–13. * $p<0.05$; ** $p<0.01$; *** $p<0.001$; **** $<0.0001$ by one-way ANOVA.

The online version of this article includes the following source data and figure supplement(s) for figure 2:

**Source data 1.** *Tfr1* myeloid conditional knockout female mice develop normally and display increased trabecular bone mass in femurs at 10-week-old and 6-month-old ages.

**Figure supplement 1.** The trabecular bone mass and structure among three different genotypes of control mice are similar.

**Figure supplement 1—source data 1.** The trabecular bone mass and structure among three different genotypes of control mice are similar.

**Figure supplement 2.** Partial deletion of Tfr1 by one-allele of LysM-Cre has no effects on bone mass and structure in male and female mice on C57BL6/J background.

**Figure supplement 2—source data 1.** Partial deletion of Tfr1 by one-allele of LysM-Cre has no effects on bone mass and structure in male and female mice on C57BL6/J background.

**Figure supplement 3.** Loss of Tfr1 in myeloid osteoclast precursor cells did not alter the serum iron and hepcidin levels and had no effects on erythropoiesis in 10-week-old male and female mice.

**Figure supplement 3—source data 1.** Loss of Tfr1 in myeloid osteoclast precursor cells did not alter the serum iron and hepcidin levels and had no effects on erythropoiesis in 10-week-old male and female mice.

**Figure supplement 4.** *Tfr1* myeloid conditional knockout male mice develop normally and exhibit no significant changes in trabecular and cortical bone.

**Figure supplement 4—source data 1.** *Tfr1* myeloid conditional knockout male mice develop normally and exhibit no significant changes in trabecular and cortical bone.

**Figure supplement 5.** Loss of Tfr1 in myeloid osteoclast precursor cells had less effects on trabecular bone mass in lumber vertebrae than femurs.

**Figure supplement 5—source data 1.** Loss of Tfr1 in myeloid osteoclast precursor cells had less effects on trabecular bone mass in lumber vertebrae than femurs.

**Figure supplement 6.** Deletion of *Tfr1* in cathepsin K-Cre expressing osteoclasts results in increased trabecular bone mass in distal femurs of 10-week-old male and female mice.

*Figure 2 continued on next page*

*Figure 2 continued*

**Figure supplement 6—source data 1.** Deletion of *Tfr1* in cathepsin K-Cre expressing osteoclasts results in increased trabecular bone mass in distal femurs of 10-week-old male and female mice.

**Figure supplement 7.** Deletion of Tfr1 in cathepsin K-Cre expressing osteoclasts slightly increases trabecular thickness of lumber vertebrae of 10-week-old male and female mice.

**Figure supplement 7—source data 1.** Deletion of Tfr1 in cathepsin K-Cre expressing osteoclasts slightly increases trabecular thickness of lumber vertebrae of 10-week-old male and female mice.

**Figure supplement 8.** Estrogen but not androgen synergistically with Tfr1-deficiency inhibits the mRNA expression of mitochondrial cytochrome c oxidase subunit I in osteoclast lineage cells.

**Figure supplement 8—source data 1.** Estrogen but not androgen synergistically with Tfr1-deficiency inhibits the mRNA expression of mitochondrial cytochrome c oxidase subunit I in osteoclast lineage cells.

**Figure supplement 9.** The deletion efficiency of Tfr1 by LysM-Cre is more potent in the distal femur than in the lumber vertebra as examined by quantitative PCR.

**Figure supplement 9—source data 1.** The deletion efficiency of Tfr1 by LysM-Cre is more potent in the distal femur than in the lumber vertebra as examined by quantitative PCR.

increase in female but not male Tfr1$^{\Delta lysM}$ mice (data not shown). By contrast, a moderate increase in trabecular bone mass of vertebral bones was observed in 6-month-old, but not 10-week-old, Tfr1$^{\Delta lysM}$ female mice (*Figure 2—figure supplement 5*). Again, the vertebral bone mass of 6-month-old control and Tfr1$^{\Delta lysM}$ male mice was indistinguishable (data not shown). Taken together, these results indicate that the effects of Tfr1-deletion in osteoclast precursor cells on trabecular bone mass in post-pubertal and adult mice are gender- and site-dependent. To determine if loss of Tfr1 in osteoclasts affects ovariectomy (OVX)-induced bone loss, control and Tfr1$^{\Delta lysM}$ female mice at 4.5 months of age were ovariectomized to mimic human post-menopausal osteoporosis and skeletal phenotype was determined at 6 months of age. While OVX induced bone loss in Tfr1$^{\Delta lysM}$ mice, the trabecular bone mass of Tfr1$^{\Delta lysM}$ mice was significantly higher than control mice after OVX.

To determine the cellular processes that contribute to increased trabecular bone mass caused by deletion of Tfr1 in myeloid cells in mice, we performed histology and histomorphometry analysis of distal femurs from 10-week-old male and female Tfr1$^{\Delta lysM}$ mice and their corresponding controls. Consistent with the µCT data, fast green and TRAP (tartrate-resistant acid phosphatase) staining of decalcified paraffin-embedded sections (*Figure 3A* and *Figure 3—figure supplement 1A*) demonstrated increased trabecular mass, Tb.N, Tb.Th, and decreased Tb.Sp in female but not male Tfr1$^{\Delta lysM}$ mice compared to their control littermates (*Figure 3B-E* and *Figure 3—figure supplement 1B–1E*). While the TRAP stained osteoclast surface adjusted by bone surface (Oc.S/BS), a histological parameter for osteoclast density, was the same in Tfr1$^{\Delta lysM}$ mice of both genders as in control mice (*Figure 3F* and *Figure 3—figure supplement 1F*), the total Oc.S which reflects the number of osteoclasts was increased in female Tfr1$^{\Delta lysM}$ but not male mice due to expanded trabecular bone mass in female knockout mice (*Figure 3G* and *Figure 3—figure supplement 1G*). The increased number of osteoclasts in distal femurs of Tfr1$^{\Delta lysM}$ female mice compared to control female mice was further supported by a >twofold higher level of mRNAs of osteoclast marker genes, *Acp5* (encoding TRAP) as measured by quantitative PCR (qPCR) (*Figure 3—figure supplement 2*). However, serum levels of bone resorption markers, TRAP5b and CTx-I, were similar between the control and Tfr1 deficient female mice (*Figure 3H-I* and *Figure 3—figure supplement 1H–1I*), thus reflecting a likely decrease in the activity of individual osteoclasts in female Tfr1$^{\Delta lysM}$ mice in vivo. Further experiments are needed to confirm this assertion.

Since osteoclast-mediated bone resorption and osteoblast-mediated bone formation are coupled and regulated from each other, we also examined bone formation activities of osteoblasts in control and Tfr1$^{\Delta lysM}$ mice by dynamic bone histomorphometry analysis of tetracycline-labeled plastic-embedded tissue sections. As shown in *Figure 3J-K* and *Figure 3—figure supplement 1J–1K*, bone formation rate and osteoblast mineral apposition rate in trabecular bone were similar between control and Tfr1$^{\Delta lysM}$ male and female mice. The single tetracycline-labeled surface (sL.Pm) in female but not male Tfr1$^{\Delta lysM}$ mice was higher than their control mice, likely due to increased trabecular bone mass/surface (*Figure 3L* and *Figure 3—figure supplement 1L*). The increased serum level of P1NP (*Figure 3M* and *Figure 3—figure supplement 1M*), a systemic bone formation marker, in female but

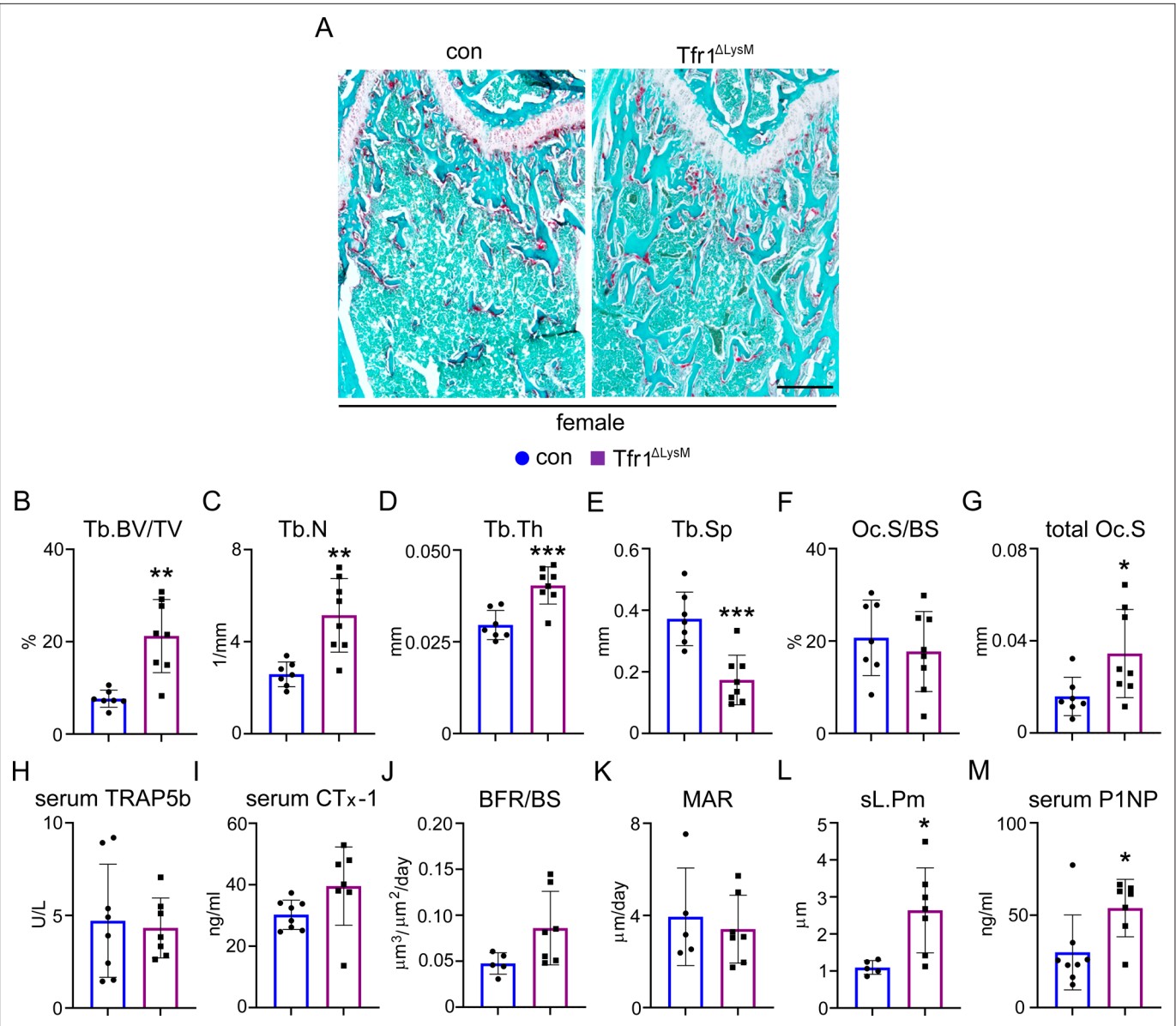

**Figure 3.** Loss of *Tfr1* in myeloid osteoclast precursors has no impacts on osteoclast number and bone formation in 10-week-old female mice. (**A**) Images of fast green and TRAP staining (scale bar = 200 μm) and (**B–G**) histomorphometric analysis of the metaphysis of decalcified distal femur histological sections of 10-week-old female control (con) and Tfr1^ΔLysM mice. (**H**), (**I**), and (**M**) quantitative measurements of serum markers for bone resorption and bone formation by ELISA. (**J–L**) Dynamic histomorphometry analysis of tetracycline-labeled sections from undecalcified distal femurs. Tb: trabecular bone; BV/TV: bone volume/tissue volume; Tb.N: trabecular number; Tb.Th: trabecular thickness; Tb.Sp: trabecular spacing; Oc.S/BS: osteoclast surface/bone surface; BFR: bone formation rate; MAR: mineral apposition rate; sL.Pm: single tetracycline labeled surface. The data are presented as mean ± SD. n=5–13. * $p<0.05$, ** $p<0.01$, and *** $p<0.001$ vs con by one-way ANOVA.

The online version of this article includes the following source data and figure supplement(s) for figure 3:

**Source data 1.** Loss of *Tfr1* in myeloid osteoclast precursors has no impacts on osteoclast number and bone formation in 10-week-old female mice.

**Figure supplement 1.** Loss of *Tfr1* in myeloid osteoclast precursors has no impacts on osteoclast number and bone formation in 10-week-old male mice.

**Figure supplement 1—source data 1.** Loss of *Tfr1* in myeloid osteoclast precursors has no impacts on osteoclast number and bone formation in 10-week-old male mice.

**Figure supplement 2.** The mRNA levels of osteoclast marker gene *Acp5* and osteoblast marker gene *Runx2* increase in the distal femurs of female Tfr1^ΔLysM mice.

**Figure supplement 2—source data 1.** The mRNA levels of osteoclast marker gene *Acp5* and osteoblast marker gene *Runx2* increase in the distal

*Figure 3 continued on next page*

Figure 3 continued

femurs of female Tfr1$^{\Delta LysM}$ mice.

**Figure supplement 3.** Deletion of *Tfr1* in cathepsin K-Cre expressing osteoclasts increases trabecular bone mass without influence on osteoclastogenesis in vivo.

**Figure supplement 3—source data 1.** Deletion of *Tfr1* in cathepsin K-Cre expressing osteoclasts increases trabecular bone mass without influence on osteoclastogenesis in vivo.

not male Tfr1$^{\Delta lysM}$ mice may be caused by increased trabecular bone and total number of osteoblasts in these mice. Accordingly, expression level osteoblast marker gene *Runx2* was increased in the distal femurs of Tfr1$^{\Delta lysM}$ female mice relative to their controls (*Figure 3—figure supplement 2*). Based on what is known about osteoclast-mediated coupling of bone formation, one would predict that the reduced osteoclast activity in Tfr1$^{\Delta lysM}$ mice should reduce bone formation. By contrast, the increased serum PINP (Procollagen 1 N-Terminal Propeptide) levels and Runx2 expression in these mice suggest increased bone formation. Thus, we do not have an explanation at present if the increased bone formation is a direct consequence of inhibition of osteoclast activity or by some other mechanisms. Further studies are needed to address this issue.

## Deletion of Tfr1 in cathepsin K-Cre expressing differentiated osteoclasts increases trabecular bone mass of long bones in both male and female mice

To determine the role of Tfr1 in differentiated osteoclasts during bone remodeling, we crossed *Tfrc*-flox mice with *Ctsk*-Cre knock-in mice, a Cre-driver that has been extensively used to disrupt genes of interest in differentiated osteoclasts (*Fujiwara et al., 2016*; *Wang et al., 2018*; *Nakamura et al., 2007*). The progeny of *Tfrc*$^{+/+}$;*Ctsk*$^{Cre/+}$ and *Tfrc*$^{flox/flox}$;Ctsk$^{Cre/+}$ congenic mice on a C57BL6/129 mixed background was used as control and osteoclast conditional knockout (Tfr1$^{\Delta CTSK}$) mice, respectively. Both male and female Tfr1$^{\Delta CTSK}$ mice were born and developed normally. The body size and weight of 10-week-old male and female Tfr1$^{\Delta CTSK}$ mice were similar to their littermate gender matched control mice (data not shown). The µCT examination of distal femurs uncovered a more than twofold increase of trabecular BV/TV with increased Tb.N, thickness, and decreased Tb.Sp in both male and female Tfr1$^{\Delta CTSK}$ mice compared to their littermate controls (*Figure 2—figure supplement 6A–6E*). A slight increase in Cort.Th was only observed in female Tfr1$^{\Delta CTSK}$ mice (*Figure 2—figure supplement 6F*). Similar to Tfr1$^{\Delta lysM}$ mice, loss of Tfr1 in differentiated osteoclasts had little effects on trabecular bone mass and structure except an increase in Tb.Th in vertebral bones of male and female Tfr1$^{\Delta CTSK}$ mice (*Figure 2—figure supplement 7*).

Histology and histomorphometry analysis of paraffin-embedded, TRAP-stained femoral sections demonstrated dramatic increase in trabecular BV/TV in both male and female Tfr1$^{\Delta CTSK}$ mice relative to their control mice (*Figure 3—figure supplement 3A and B*). These data are consistent with the µCT results shown in *Figure 2—figure supplement 6*. The number of TRAP-positive osteoclasts per bone perimeter as well as the percentage of osteoclast-covered bone surface in Tfr1$^{\Delta CTSK}$ mice of either genders were not different from their respective control mice (*Figure 3—figure supplement 3C–3D*). These results indicate that loss of Tfr1 in *Ctsk*-Cre expressing osteoclasts increases trabecular bone mass but has no effects on osteoclastogenesis in vivo.

## Tfr1-mediated iron uptake plays an important role in mature osteoclast actin cytoskeletal organization and bone resorption in vitro

To further identify the cellular and molecular mechanisms by which Tfr1 and Tfr1-mediated iron uptake regulate osteoclasts, we isolated bone marrow cells from female and male control and Tfr1$^{\Delta lysM}$ mice and cultured them with M-CSF plus RANKL for 4 days to induce osteoclast differentiation in vitro. The total number of TRAP$^+$ multinucleated osteoclasts (with ≥3 nuclei per cell) in Tfr1$^{\Delta lysM}$ cultures of both genders was not different from their respective controls. However, the Tfr1-deficient female and male osteoclasts displayed a dramatic defect in mature osteoclast spreading (*Figure 4* and *Figure 4—figure supplement 1A*). To further corroborate the effects of Tfr1-deletion on osteoclast differentiation, we measured the mRNA expression of the osteoclast marker genes *Acp5*, *Ctsk*, the master transcription factor of osteoclast differentiation *Nfatc1*, and the osteoclast fusion regulator *Dcstamp*

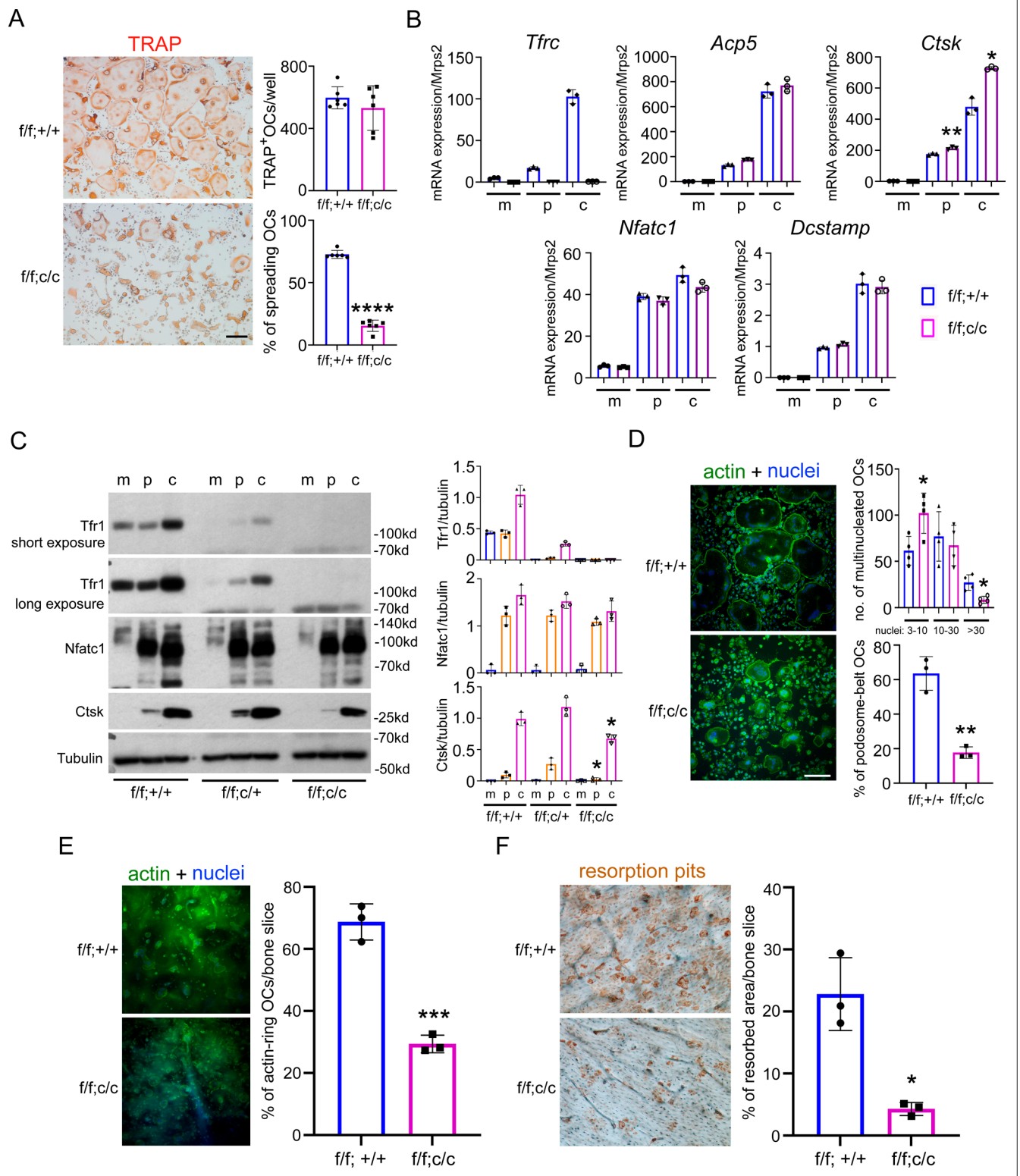

**Figure 4.** Deletion of Tfr1 in osteoclast myeloid precursors has no effects on osteoclast differentiation but attenuates cytoskeleton organization and bone resorption in mature osteoclasts cultured from female mice. (**A**) TRAP staining (scale bar = 10 µm) and quantification of numbers of total and spreading osteoclasts (OCs) cultured on plastic 24-well dishes. n=6. (**B**) Quantitative PCR detection of the mRNA expression of Tfr1 and osteoclast marker genes relative to the mitochondrial gene *Mrps2*. *Tfrc*, encoding Tfr1; *Acp5*, encoding TRAP; *Ctsk*, encoding Cathepsin K; *Nfatc1*, encoding NFATc1; *Dcstamp*, encoding DC-Stamp. n=3. (**C**) The protein level of Tfr1 and osteoclast markers, NFATc1 and Cathepsin K (Ctsk), in bone marrow

*Figure 4 continued on next page*

*Figure 4 continued*

monocytes (**m**), mono-nuclear pre-osteoclasts (**p**), and mature osteoclasts (**c**) was detected by western blotting and quantified by densitometry using the NIH ImageJ software. Tubulin served as loading control. n=3. (**D**) The actin filaments and nuclear were stained by Alexa-488 conjugated Phalloidin and Hoechst 33258, respectively, in osteoclasts cultured on glass coverslips (scale bar = 10 µm). The number of osteoclasts with different nuclei and the percentage of spreading osteoclasts with the peripheral distributed podosome-belt were counted. n=3–4. (**E**) The actin filaments and nuclear were stained by Alexa-488 conjugated Phalloidin and Hoechst 33342, respectively, in osteoclasts cultured on cortical bovine bone slices (scale bar = 10 µm). The total number of osteoclasts and the number of osteoclasts with actin-ring were counted, and the percentage of active osteoclasts per bone slices was calculated. n=3. (**F**) Resorption pits were stained by horseradish peroxidase conjugated wheat-germ agglutinin (scale bar = 20 µm). The percentage of resorbed area per bone slice was calculated using the NIH ImageJ software. The data are presented as mean ± SD. n=3. * $p<0.05$; ** $p<0.01$; *** $p<0.001$;**** $p<0.0001$ vs control (f/f;+/+) by one-way ANOVA and Student's t-test.

The online version of this article includes the following source data and figure supplement(s) for figure 4:

**Source data 1.** Deletion of Tfr1 in osteoclast myeloid precursors has no effects on osteoclast differentiation but attenuates cytoskeleton organization and bone resorption in mature osteoclasts cultured from female mice.

**Source data 2.** Immunoblotting for *Figure 4C*.

**Figure supplement 1.** Deletion of Tfr1 in osteoclast myeloid precursors has no effects on osteoclast differentiation but attenuates cytoskeleton organization and bone resorption in mature osteoclasts cultured from male mice.

**Figure supplement 1—source data 1.** Deletion of Tfr1 in osteoclast myeloid precursors has no effects on osteoclast differentiation but attenuates cytoskeleton organization and bone resorption in mature osteoclasts cultured from male mice.

**Figure supplement 1—source data 2.** Immunoblotting for *Figure 4—figure supplement 1B*.

(encoding DC-Stamp) in female control and Tfr1$^{\Delta lysM}$ BMM, mononuclear pre-osteoclasts, and multi-nucleated mature osteoclasts by qPCR. As shown in *Figure 4B*, the mRNA levels of *Acp5*, *Nfatc1*, and *Dcstamp* in Tfr1$^{\Delta lysM}$ in osteoclast precursors as well as mature osteoclasts were similar to those of respective control cells. The mRNA expression of Ctsk in Tfr1$^{\Delta lysM}$ pre-osteoclasts and mature osteoclasts was slightly higher than in control cells.

Next, we examined the protein levels of Tfr1, Nfatc1, and Ctsk in control and Tfr1-deficient osteoclast lineage cells by immunoblotting and quantified the specific bands of these osteoclast markers by densitometry using NIH Image J software. As demonstrated in *Figure 4C* and *Figure 4—figure supplement 1B*, deletion of Tfr1 by two copies of *Lyz2*-Cre (f/f;c/c) in both female and male osteoclast lineage cells had no effects on the protein level of Nfatc1 in osteoclast precursor and mature cells, whereas the level of CtsK in Tfr1$^{\Delta lysM}$ pre- and mature-osteoclasts was slightly decreased compared to control (f/f;+/+) and Tfr1 heterozygous (f/f;c/+) osteoclasts. The decreased Ctsk protein level in Tfr1$^{\Delta lysM}$ osteoclasts might contribute to impaired bone resorption in these cells as shown in *Figure 4F*. To characterize at which stage the Tfr1-deletion affects Ctsk expression and osteoclastogenesis, we counted the number of osteoclasts with 3–10, 10–30, and >30 nuclei cultured from female mice, respectively. As shown in the upper-right panel of *Figure 4D*, the number of osteoclasts with 3–10 nuclei in Tfr1$^{\Delta lysM}$ culture was higher than control. While there was no difference in the number of osteoclasts with 10–30 nuclei between control and Tfr1$^{\Delta lysM}$ cultures, the number of large osteoclasts with >30 nuclei in Tfr1$^{\Delta lysM}$ culture was lower than control. However, the mRNA expression of osteoclast fusion gene, *Dcstamp,* was similar in control and Tfr1$^{\Delta lysM}$ osteoclasts. The exact mechanism(s) of decreased fusion in the late stage of osteoclastogenesis in Tfr1$^{\Delta lysM}$ cultures need further investigation in future.

Since actin cytoskeleton organization plays an essential role in osteoclast spreading and the formation of podosome-belts in osteoclasts cultured on plastic dishes and actin-rings in osteoclasts cultured on bone matrix, we then stained actin filaments with Alexa-488 conjugated Phalloidin in control and Tfr1-deficient female and male osteoclasts cultured on plastic and bone. While the podosome-belts and actin-rings formed normally in control osteoclasts, these actin-based structures were dysregulated in Tfr1-deficient female and male osteoclasts (*Figure 4D-E* and *Figure 4—figure supplement 1C–1D*). Since formation of actin-rings is a hallmark of osteoclast activation and function, these results indicate that loss of Tfr1 in osteoclast suppresses osteoclast function. Supporting this hypothesis, we found that Tfr1-deficient osteoclasts resorbed less bone than controls as revealed by the resorption-pit staining on cortical bovine bone slices (*Figure 4F* and *Figure 4—figure supplement 1E*). Tfr1$^{\Delta Ctsk}$ osteoclasts displayed similar in vitro phenotype as Tfr1$^{\Delta lysM}$ osteoclasts (data not shown).

Besides regulating iron-uptake, Tfr1 also plays non-canonical roles in regulation of intestinal epithelial homeostasis and mitochondrial morphology and function (*Chen et al., 2015*; *Senyilmaz*

et al., 2015). To determine whether the defects in cytoskeleton organization and resorptive function of Tfr1-deficient osteoclasts were caused by low intracellular iron content, we treated control and Tfr1$^{\Delta lysM}$ osteoclast cultures with increased doses of hemin (ferric chloride heme) and ferric ammonium citrate (FAC), which are transported into cells via heme transporters Hcp1 (encoded by *Slc46a1*) and NTBI transporter Zip14 (encoded by *Slc39a14*), respectively (*Figure 1—figure supplement 1*). While 10 µM of hemin slightly stimulated osteoclast spreading, cytoskeleton organization, and bone resorption in control osteoclasts, hemin dose-dependently increased the number of spreading osteoclasts in Tfr1-deficient osteoclasts (*Figure 5A*). In contrast, FAC had no effects on control or Tfr1-deficient osteoclasts (*Figure 5B*). At 10 µM concentration, hemin significantly rescued the cytoskeleton organization and bone resorption defects of Tfr1-depleted osteoclasts (*Figure 5C-E*). These results also suggest that heme-uptake, but not NTBI and ferritin pathways, is an alternative iron acquiring route in osteoclasts in the absence of Tfr1. Therefore, both our in vivo and in vitro findings unveil a critical role of Tfr1-mediated cellular iron homeostasis in osteoclast activation/function. It should be noted that even high concentration (10 µM) of hemin can only partially rescue the defects in osteoclast spreading and cytoskeleton organization (*Figure 5A, C and D*), indicating that osteoclasts have limited alternative iron acquisition pathways in the absence of Tfr1 as in other cells.

## Depletion of Tfr1 in osteoclast lineage cells attenuates mitochondrial biogenesis, ROS production, and OXPHOS predominantly in mature osteoclasts

To determine which intracellular pathways are most affected by loss of Tfr1 in osteoclast lineage cells, we did quantitative proteomic analysis to identify proteins that are up- or downregulated in Tfr1-deficient BMM, mononuclear osteoclast precursors, and multinucleated mature osteoclasts. The heatmaps were generated using the 'heatmap' function in the R-package (*Figure 6A*). By a 1.2-log2fold cut-off, we found that a total of 135, 497, and 2562 proteins were influenced by loss of Tfr1 in monocytes, pre-osteoclasts, and mature osteoclasts, respectively (*Supplementary files 1–3*). The abundance of proteins involved in Tf-dependent iron uptake pathway, including Tfr1, Steap3/4, and DMT1, was all decreased in Tfr1-deficient cells (*Supplementary files 1–3*). The greater number of affected proteins in mature osteoclasts correlated with a high level of Tfr1 and Tfr1-mediated iron uptake in osteoclasts (*Figure 1*). By Ingenuity pathway analysis (IPA) analysis, we identified that the mitochondrial OXPHOS pathway and sirtuin signaling pathway were the most significantly down- and upregulated pathways in mature osteoclasts, respectively (*Figure 6B* and *Supplementary file 4*). Among the mitochondrial proteins affected by loss of Tfr1 and Tf-dependent iron uptake in osteoclasts, the levels of proteins functioning in mitochondrial respiratory complex I to complex III decreased, whereas the proteins participating in complex V, that lack heme and Fe-S clusters (*Figure 1—figure supplement 1*), were increased probably by a compensate mechanism. The proteins in the complex IV were not affected by iron-deficiency (*Figure 6C* and *Supplementary file 4*). To consolidate the finding from quantitative proteomic results that cellular iron regulates the components of mitochondrial respiration chain, we did immunoblotting to detect the changes of mitochondrial respiratory complexes in control and Tfr1$^{\Delta lysM}$ monocytes and mature osteoclasts treated with either vehicle or 10 µM hemin using a cocktail of antibodies recognizing the representative components of mitochondrial respiratory complexes I–V. As shown in *Figure 6D* and *Figure 6—figure supplement 1*, there was an obvious decrease in the mitochondrial complexes II and III in Tfr1$^{\Delta lysM}$ monocytes and osteoclasts compared to corresponding control cells. Hemin stimulated the mitochondrial complexes I and II in control and complexes I to IV in Tfr1$^{\Delta lysM}$ osteoclasts, indicating that Tfr1-mediated iron uptake plays an important role in osteoclast mitochondrial OXPHOS pathway. Although *Figure 6D* provides complementary evidence for proteomic data shown in *Figure 6B, C*, it should be pointed out that the antibody cocktail used in this immunoblotting contains antibodies against only one component of each of five mitochondrial respiratory complexes. Thus, the result of *Figure 6D* could not reflect the complete changes of mitochondria in Tfr1-deficient osteoclast lineage cells.

To further characterize how iron-deficiency modulates mitochondrial biogenesis and metabolism in osteoclast lineage cells, we first stained the control and Tfr1$^{\Delta lysM}$ osteoclast lineage cells with fluorescence-labeled Mito Tracker. Manual quantification of Mito Tracker fluorescent intensity of individual cells showed a small decrease in mitochondrial mass in Tfr1-deficient mature osteoclasts but not in monocytes and pre-osteoclasts (*Figure 7A*). Next, we examined the mitochondrion-derived

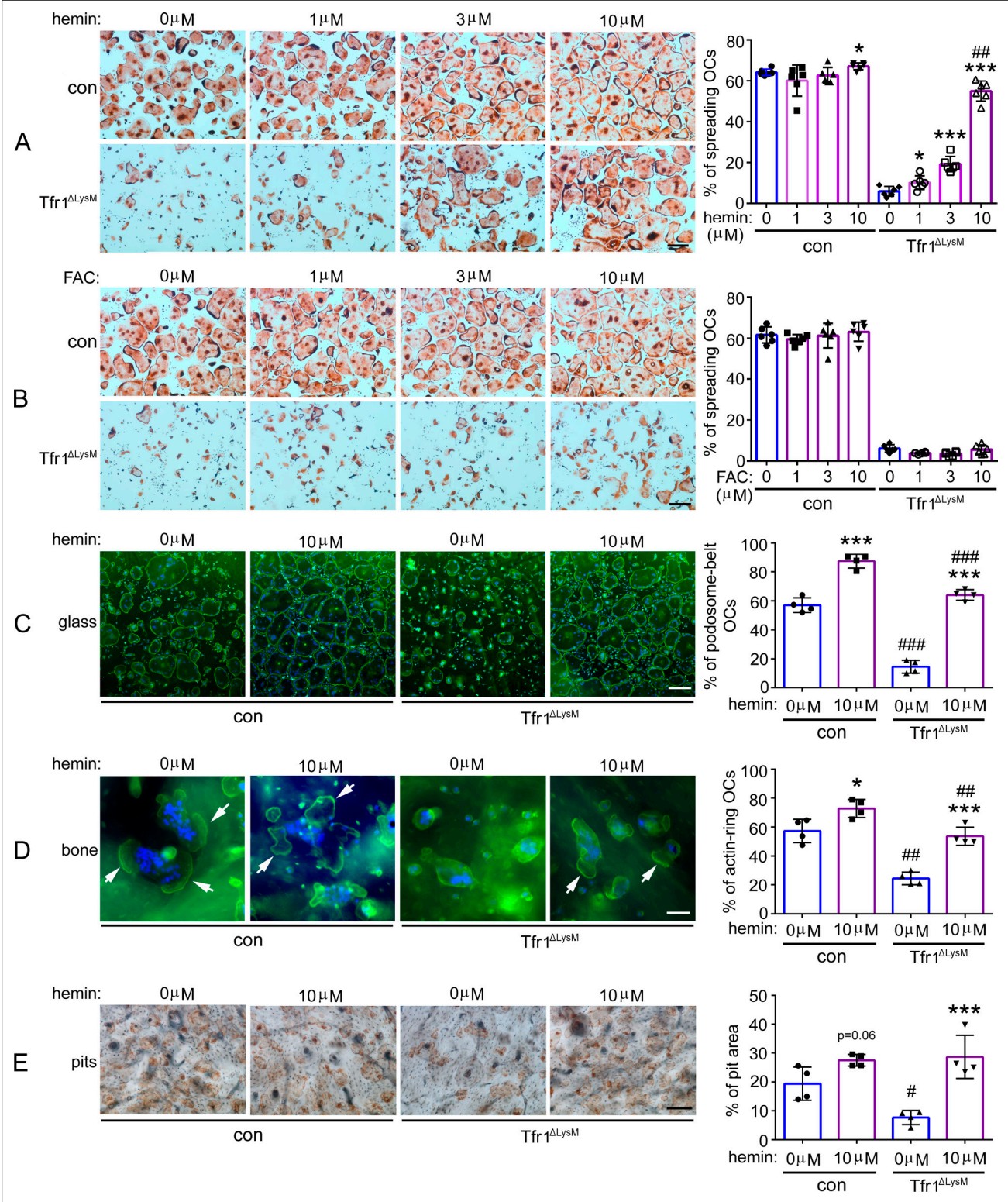

**Figure 5.** High dose of hemin but not ferric ammonium citrate rescues the phenotypes of Tfr1-deficient osteoclasts. (**A**) and (**B**) TRAP staining and quantification of the number of spreading osteoclasts in control (con) and Tfr1$^{\Delta LysM}$ osteoclast cultures. n=6. scale bar = 10 µm. (**C**) and (**D**) Staining of actin filaments and nuclear and quantification of the number of podosome-belt and actin-ring osteoclasts cultured on glass coverslips and bone slices, respectively. n=4. scale bar = 10 µm. (**E**) Resorption pit staining in con and Tfr1$^{\Delta LysM}$ cultures. Scale bar = 20 µm. The data are presented as mean ± SD. n=4. * p<0.05 and *** p<0.001 vs vehicle-treated cells; # p<0.05, ## p<0.01, and ### p<0.001 vs vehicle- or 10 µM hemin-treated control cells, respectively, by one-way ANOVA.

*Figure 5 continued on next page*

*Figure 5 continued*

The online version of this article includes the following source data for figure 5:

**Source data 1.** High dose of hemin but not ferric ammonium citrate rescues the phenotypes of Tfr1-deficient osteoclasts.

ROS by MitoSOX-staining of control and Tfr1$^{\Delta lysM}$ cultured cells. As demonstrated in *Figure 7B*, loss of Tfr1 significantly decreased mitochondrial ROS production in monocytes and osteoclasts which was more pronounced in mature OCs. Insufficient iron in Tfr1$^{\Delta lysM}$ mature osteoclasts also resulted in a mild decrease in mitochondrial membrane potential (*Figure 7C*).

Lastly, we measured mitochondrial respiration using the Seahorse Extracellular Flux Analyzer in control and Tfr1-deficient osteoclast lineage cells isolated. As depicted in *Figure 8A*, the basal mitochondrial respiration is the direct measure of oxygen consumption rate (OCR) attributed to the mitochondrial electron transport chain (ETC). The maximum respiration is induced by incubating cells with the uncoupler, carbonyl cyanide 4-(trifluoromethoxy)phenylhydrazone (FCCP). The ATP-linked respiration is measured by exposing cells to oligomycin, an inhibitor of ATP synthase complex V. The non-mitochondrial respiration, which is the remaining OCR when the ETC activity is completely abolished by a mixture of rotenone/antimycin A. This parameter reflects the portion of cellular respiration by oxygen-consuming enzymes such as NADPH (nicotinamide adenine dinucleotide phosphate) oxidases, heme oxygenases, and/or lipoxygenases. The residual OCR after inhibition by oligomycin is the measure of the protons pumped through ETC, which consumes oxygen without generating any ATP (due to inhibited ATP synthase activity by the drug) and is referred to the H$^{+}$ leak. As shown in *Figure 8B–H*, all these mitochondrial respiratory parameters increased dramatically in control and TfR1-null mature osteoclasts compared to their respective precursors. However, there were >twofold decreases of these measures in Tfr1-deficient mature osteoclasts relative to control osteoclasts. Furthermore, we calculated the reserve respiratory capacity, which is the difference between the maximum and basal respiration that can be utilized in the event of a sudden energy demand in cells. We reported this as a percentage for each group of cells by using the formula of ([maximum OCR - basal OCR]/maximum OCR ×100). Again, the reserve respiratory capacity in Tfr1-null osteoclasts was greatly reduced when compared to control cells (*Figure 8E*). By contrast, there was no difference in the coupling efficiency (the ratio of ATP-linked OCR to basal OCR) of control and Tfr1$^{\Delta lysM}$ osteoclast lineage cells (*Figure 8G*).

## TfR1-mediated iron uptake modulates osteoclast actin cytoskeleton through the WAVE regulatory complex (WRC)

To identify the mechanisms by which cellular iron and energy metabolism regulate osteoclast cytoskeleton, we turned to analyze the level of key osteoclast cytoskeleton regulating proteins (*Blangy et al., 2020*) in Tfr1-deficient osteoclast lineage cells relative to their control cells in our quantitative proteomic data bases (*Supplementary files 1–3*). More cytoskeleton proteins in mature osteoclasts were affected by Tfr1-dificiency than those in osteoclast precursors (*Supplementary file 5*). Among them, 25 cytoskeleton proteins were downregulated, whereas 17 proteins were upregulated by more than 1.2-log2fold in Tfr1-dificient osteoclasts compared to control cells. 15 osteoclast cytoskeleton-regulatory proteins remained the same between control and Tfr1-dificient osteoclasts. IPA analysis of the cytoskeleton-regulating pathways affected by Tfr1-deficiency in osteoclasts (*Figure 9A*) unveiled that the protein level of several molecules involved in activation of β$_3$-integrin pathway, which is critical for osteoclast cytoskeleton organization and bone resorption (*Teitelbaum, 2011*), was decreased in Tfr1-null osteoclasts including β$_3$-integrin (Itgb3), c-Src, Rap1a/b, Rap2b, Rapgef1, and Rap1gds (*Supplementary file 5*). A few actin-bundling and regulating proteins such as myosin heavy chain 14, α-actinin 1 and 4 (Actn1 and Actn4), and filamin A (Flna) were also downregulated in Tfr1-depleted osteoclasts. The branching and polymerization of filamentous actin in eukaryotic cells is mediated by the Arp2/3 complex which is controlled by the WRC. Mammalian WRC contains five subunits of Cyfip1/2, Hem1/2 (encoded by *Nckap1l* and *Nckap1*, respectively), Abi1/2/3, HSPC300, and WAVE1/2/3 (*Rottner et al., 2021*). The WRC is activated by the small GTPases Rac1 and Arf1 through their direct binding to Cyfip1/2 and Hem1/2. Strikingly, 12 out of 25 downregulated cytoskeletal proteins in Tfr1-null osteoclasts (*Supplementary file 5*) are the components of the Rac1/Arf-WRC-Arp2/3 axis. These data indicate that Tfr1-mediated iron uptake regulates osteoclast actin

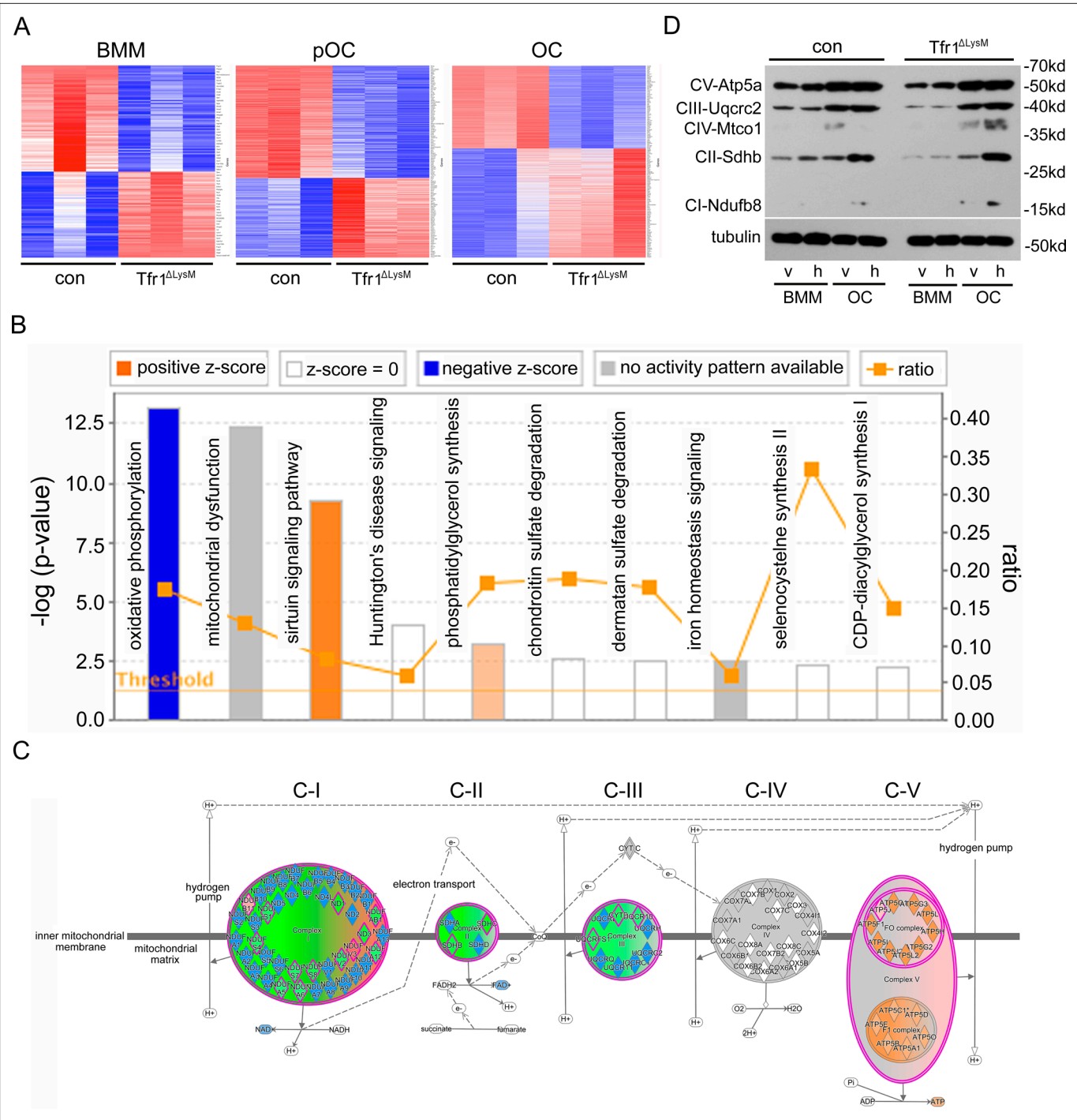

**Figure 6.** The mitochondrial oxidative phosphorylation is mostly affected by Tfr1-deficiency in mature osteoclasts. (**A**) Heatmaps of proteins that are differentially regulated in control (con) and Tfr1$^{\Delta LysM}$ bone marrow monocytes (BMM), mononuclear pre-osteoclasts (pOC), and mature osteoclasts (OC) identified by quantitative proteomics. (**B**) The signaling pathways that are affected by Tfr1-deficiency in mature osteoclasts identified by Ingenuity pathway analysis (IPA). (**C**) The changes of proteins along the mitochondrial respiration chain that are regulated by Tfr1 in mature osteoclasts. C-I to C-V: mitochondrial respiratory complex-I to complex-V. (**D**) Western blotting detection of the components of mitochondrial respiratory C-I to C-V in vehicle and 10 μM hemin treated con and Tfr1$^{\Delta LysM}$ BMM and OCs.

The online version of this article includes the following source data and figure supplement(s) for figure 6:

**Source data 1.** Immunoblotting for *Figure 6D*.

**Figure supplement 1.** Quantification of immunoblotting bands in *Figure 6D*.

**Figure supplement 1—source data 1.** Quantification of immunoblotting bands in *Figure 6D*.

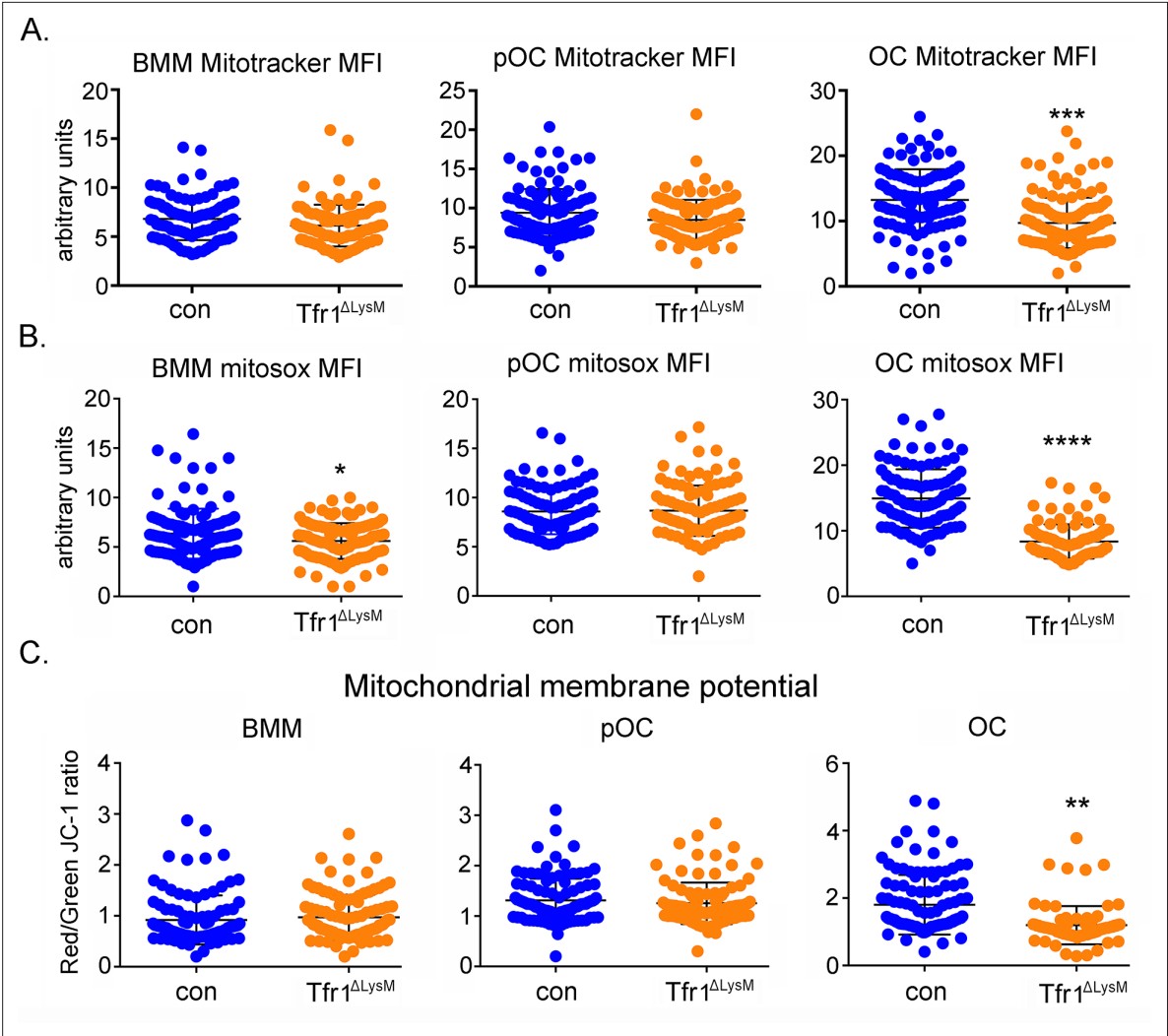

**Figure 7.** Loss of Tfr1 inhibits mitochondrial mass, reactive oxygen species (ROS) production, and membrane potential in mature osteoclasts. (**A**) Quantification of mitochondrial mass by Mito Tracker green staining in control (con) and Tfr1$^{\Delta LysM}$ bone marrow monocytes (BMM), mononuclear pre-osteoclasts (pOC), and mature osteoclasts (OC). (**B**) Measurement of mitochondria-derived ROS by Mitosox staining in con and Tfr1-deficient osteoclast lineage cells. (**C**) Measurement of mitochondrial membrane potential by JC-1 cationic carbocyanine dye staining in con and cKO osteoclast lineage cells. The data are presented as mean ± SD. n=25–50. ** p<0.01, *** p<0.001, and **** p<0.0001 vs con by unpaired Student's t-test. MFI: mean fluorescent intensity per cell in arbitrary units.

The online version of this article includes the following source data for figure 7:

**Source data 1.** Loss of Tfr1 inhibits mitochondrial mass, reactive oxygen species (ROS) production, and membrane potential in mature osteoclasts.

cytoskeleton organization, at least in part, through modulating the stability of the WRC-Arp2/3 actin-regulating module.

To identify the mechanisms by which Tfr1 regulates osteoclast WRC complex and cytoskeleton, we examined the activation (phosphorylation) of protein tyrosine kinase c-Src which plays a pivotal role in osteoclast cytoskeleton organization downstream of integrin and M-CSF as well as RANKL signaling pathways in control and Tfr1$^{\Delta lysM}$ osteoclasts stimulated by M-CSF and RANKL. We also assayed the activation of small GTPases Rac1 and Arf1 that are upstream stimulators of the WRC complex in responses to M-CSF and RANKL in control and Tfr1$^{\Delta lysM}$ osteoclasts using GST-pulldown kits from Thermo-Fisher Scientific Inc. As shown in *Figure 9B* and *Figure 9—figure supplement 1*, while both M-CSF and RANKL activated c-Src in control and Tfr1$^{\Delta lysM}$ osteoclasts, the level of active (phosphory-lated) c-Src in M-CSF, but not RANKL, stimulated Tfr1$^{\Delta lysM}$ osteoclasts was lower than control cells at all time points. Similarly, M-CSF induced Rac1 activation in form of Rac1-GTP was attenuated in Tfr1$^{\Delta lysM}$

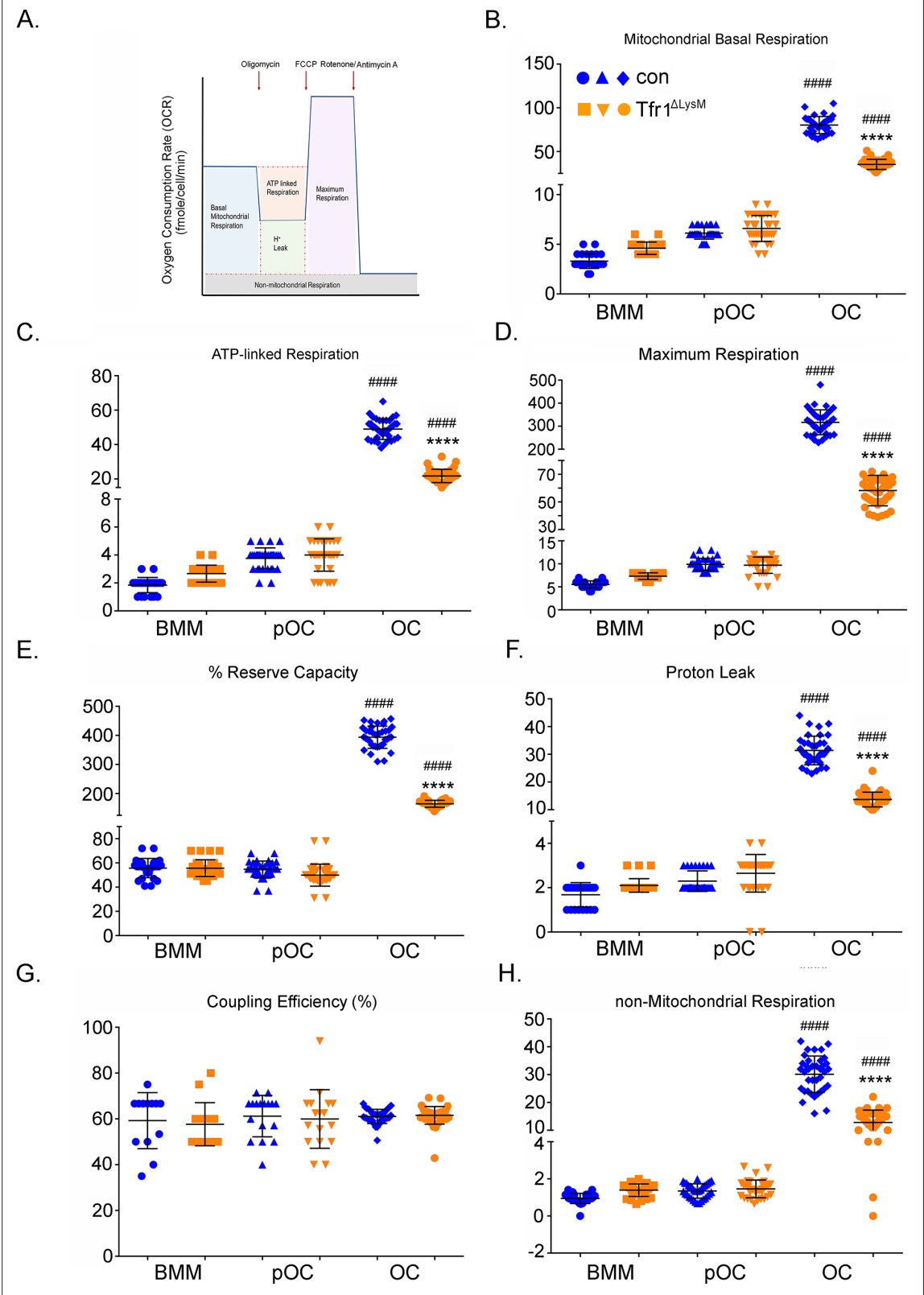

**Figure 8.** Tfr1-deletion in osteoclast lineage cells impairs mitochondrial and non-mitochondrial respirations in mature osteoclasts. (**A**) A graphic illustration of oxygen consumption measured by a Seahorse Extracellular Flux analyzer. FCCP: carbonyl cyanide p-trifluoro-methoxyphenyl hydrazone, a synthetic mitochondrial uncoupler. (**B–H**) Different fractions of mitochondrial and non-mitochondrial respirations per cell of control (blue) and Tfr1$^{\Delta LysM}$

*Figure 8 continued*

(orange) bone marrow monocytes (BMM), mononuclear pre-osteoclasts (pOC), and mature osteoclasts (OC). The data are presented as mean ± SD. n=15-30. #### p<0.0001 vs BMM and pOC; **** p<0.0001 vs control OC by unpaired Student's t-test.

The online version of this article includes the following source data for figure 8:

**Source data 1.** Tfr1-deletion in osteoclast lineage cells impairs mitochondrial and non-mitochondrial respirations in mature osteoclasts.

osteoclasts compared to control cells at all time points, whereas RANKL did not arise Rac1-GTP level in control and Tfr1$^{\Delta lysM}$ osteoclasts. In regard to Arf1, M-CSF stimulation slightly decreased Arf1-GTP in both control and Tfr1$^{\Delta lysM}$ osteoclasts. The level of Arf1-GTP in Tfr1$^{\Delta lysM}$ osteoclasts was higher than control cells after starvation and 5 min of M-CSF stimulation. RANKL induced a mild decrease of Arf1-GTP in control osteoclasts and a little increase of Arf1-GTP in Tfr1$^{\Delta lysM}$ osteoclasts verse control cells after 30 min of RANKL incubation, although the Arf1-GTP was lower in starved and 15 min stimulated Tfr1$^{\Delta lysM}$ osteoclasts than control cells. These results indicate that loss of Tfr1 in osteoclasts influences M-CSF-mediated activation of the Src-Rac1-WRC axis, leading to disorganized cytoskeleton and compromised bone resorption.

To test whether overexpression of the components of WRC complex can rescue the cytoskeleton defect in Tfr1$^{\Delta lysM}$ osteoclasts, we constructed a recombinant retroviral vector expressing murine Hem1. After retroviral transduction in control and Tfr1-deleting BMM, we confirmed the expression of recombinant Hem1 by western blotting (*Figure 9C*). The overexpression of Hem1 enhanced the number of activated osteoclasts in control osteoclasts and partially rescued the cytoskeletal defect in Tfr1$^{\Delta lysM}$ osteoclasts (*Figure 9D*), suggesting that the other component(s) of the WRC complex or distinct pathways might also be involved in the Tfr1-mediated regulation of osteoclast cytoskeleton.

## Discussion

Loss of Tfr1 in osteoclast lineage cells results in >50% decrease in the total intracellular iron content in mature osteoclasts, whereas the iron levels in monocytes and pre-osteoclasts are not affected by Tfr1-deficiency (*Figure 1*). Since the heme transporter Hcp1 is upregulated during osteoclast differentiation and the fact that hemin but not FAC (absorbed through NTBI) is able to partially rescue the Tfr1-deficient osteoclast phenotypes (*Figure 5*), the residual cellular iron in Tfr1-deficient osteoclasts could be obtained through uptake of heme. Given that the key molecules involved in Tf-dependent and heme-dependent iron uptake pathway are significantly upregulated during osteoclast differentiation and that the expressions of NTBI iron transporter Zip14 and ferritin transporter (Scara-5 and Tim-2) are either decreased in osteoclasts or undetectable in osteoclast lineage cells (*Figure 1A* and data not shown), these results indicate that mature osteoclasts acquire extracellular iron largely through uptake of Tf and heme.

Although iron chelation by desferrioxamine has been shown to inhibit osteoclasts and prevent bone loss in ovariectomized mice (*Ishii et al., 2009*) and in a mouse model of Alzheimer's disease (Guo et al.), the intrinsic functions of Tfr1 and Tfr1-mediated iron uptake in osteoclast lineage cells in vivo have not been investigated using genetically modified mouse models. To fill in this knowledge gap, we have generated two lines of *Tfrc* conditional knockout mice in which the Tfr1 expression is disrupted in *Lyz2*-Cre expressing myeloid osteoclast precursors and *Ctsk*-Cre expressing differentiated osteoclasts, respectively. The μCT analysis of control and Tfr1$^{\Delta lysM}$ mice at three different ages ranging from the pre-pubertal 3-week-old to the post-pubertal 10-week-old and the adult 6-month-old has unveiled a significant increase in trabecular bone mass in post-pubertal and adult female, but not male, Tfr1$^{\Delta lysM}$ mice compared to their age- and gender-matched control mice (*Figure 2* and *Figure 2— figure supplement 4*). Although the significant increase of trabecular bone mass was observed in both genders of 10-week-old *Tfr1$^{\Delta Ctsk}$* mice, this phenotype is more pronounced in female knockout mice (*Figure 2—figure supplement 6*). These results indicate that the accrual of trabecular bone mass in Tfr1-deficient mice is gender-dependent. The exact mechanisms underlying the gender difference in the trabecular bone phenotype of mice upon the Tfr1-deficiency in osteoclasts are unknown and need further investigation since sex as a biological variable in mice and humans has been increasingly appreciated (*Bhargava et al., 2021*). A similar finding has been recently reported in *Slc2a1* (encoding the glucose transporter protein type 1, Glut1) myeloid conditional knockout mice (*Li et al., 2020a*). While Glut1-deficient BMM cultured from both male and female knockout mice have similar degree

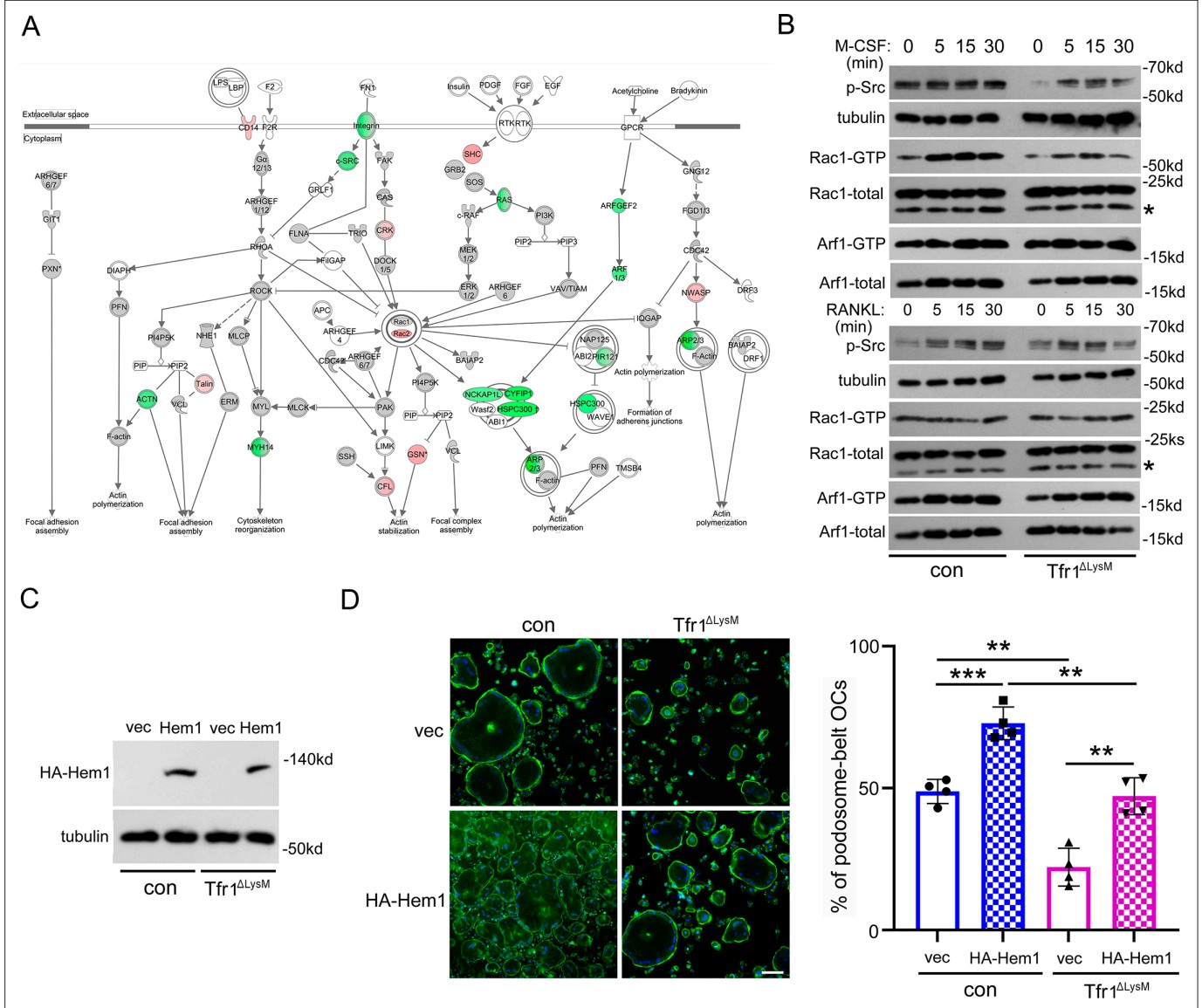

**Figure 9.** Loss of Tfr1 in osteoclasts inhibits the macrophage colony-stimulating factor (M-CSF) stimulated activation of c-Src and Rac1, and overexpression of Hem1 partially recuses the cytoskeletal organization defect in Tfr1-null osteoclasts. (**A**) A schematic map of cytoskeletal pathways generated by Ingenuity pathway analysis (IPA). The downregulated proteins are shown in green and upregulated proteins are marked in red orange. (**B**) Western blotting detection of M-CSF and receptor activator of NF-$\kappa$ B ligand (RANKL) induced phosphorylation of c-Src and the GTP-bound (active) form of small GTPase Rac1 and Arf1 in con and Tfr1^ΔLysM osteoclasts. The stars indicate unspecific bands. (**C**) Detection of HA-tagged Hem1 (encoded by Nckap1l) by western blotting in retroviral transduced con and Tfr1^ΔLysM bone marrow monocytes expressing empty vector (vec) and recombinant Hem1. Tubulin served as loading control. (**D**) The staining of actin filaments and nuclear in osteoclasts cultured on glass coverslips and quantification of the number of osteoclasts with podosome-belt. scale bar = 10 μm. The data are presented as mean ± SD. n=4. ** p<0.01 and *** p<0.001 by one-way ANOVA.

The online version of this article includes the following source data and figure supplement(s) for figure 9:

**Source data 1.** Loss of Tfr1 in osteoclasts inhibits the macrophage colony-stimulating factor (M-CSF) stimulated activation of c-Src and Rac1 and overexpression of Hem1 partially recuses the cytoskeletal organization defect in Tfr1-null osteoclasts.

**Source data 2.** Immunoblotting for *Figure 9B*.

**Figure supplement 1.** Quantification of immunoblotting bands in *Figure 9B*.

**Figure supplement 1—source data 1.** Quantification of immunoblotting bands in *Figure 9B*.

of blunted osteoclast differentiation in vitro, deletion of *Slc2a1* in *Lyz2*-Cre expressing myeloid cells in mice inhibits osteoclastogenesis and leads to increased trabecular bone mass in female but not male mice.

Estrogen has been reported to regulate systemic and cellular iron homeostasis by inhibiting the expression of hepcidin, a liver-derived iron regulating hormone that binds to and induces degradation of iron exporter Fpn (*Hou et al., 2012*; *Yang et al., 2012*; *Ikeda et al., 2012*); however, the levels of serum hepcidin and total iron are similar in control and Tfr1$^{\Delta lysM}$ male and female mice (*Figure 2—figure supplement 3*) excluding this possible mechanism. On the other hand, our in vitro supplemental experiment has demonstrated that 17-β estradiol inhibits the mRNA expression of *Tfrc* in mature osteoclasts and synergistically with Tfr1-deficiency attenuates the mRNA expression of *mt-Co1* (encoding the mitochondrial cytochrome c oxidase subunit I, mt-Co1), whereas dihydrotestosterone has no effects on Tfr1, Fpn, and mt-Co1 expression in osteoclast lineage cells (*Figure 2—figure supplement 8*). This finding is consistent with our recent published work that estrogen attenuates mitochondrial OXPHOS and ATP production in osteoclasts (*Kim et al., 2020*). The tissue- and bone cell type-specific sex differences in mitochondria and the underlying mechanisms in skeletons in both mice and humans need further investigation as recently reported in adipose tissue (*Chella Krishnan et al., 2021*).

It is well established that estrogen-mediated mechanisms play a key role in mediating bone accretion during pubertal growth period. During this developmental period, estrogen is known to exert an anabolic effect on osteoblast besides its well-established role in inhibiting osteoclast functions. However, the loss of estrogen as seen during menopause or due to ovariectomy in adults leads to bone loss primarily due to increased bone resorption. The increased trabecular bone volume of Tfr1$^{\Delta lysM}$ mice is evident at the post-pubertal age of 10 weeks but is not different at the pre-pubertal age of 3 weeks (*Figure 2*), thus suggesting that the skeletal phenotype gets manifested during pubertal and post-pubertal growth periods. While loss of Tfr1 does not affect estrogen deficiency-induced bone loss, the trabecular bone mass of Tfr1$^{\Delta lysM}$ mice is persistently higher than control mice after OVX. The issue of whether the skeletal phenotype in 10-week-old Tfr1$^{\Delta lysM}$ mice is dependent on pubertal increase in estrogen remains to be evaluated by performing OVX at pre-pubertal age of 3 weeks in future.

Both *Lyz2*-Cre and *Ctsk*-Cre mice are commonly used to conditionally delete the floxed genes at different stages of osteoclast lineage cells (*Dallas et al., 2018*). While the *Lyz2*-Cre deletes genes in monocytes of osteoclast precursor cells, it also targets other myeloid cells such as neutrophils and dendritic cells (*Abram et al., 2014*). Although Nfatc1-deletion by *Mx1*-Cre attenuates osteoclastogenesis and bone resorption in mice, Nfatc1$^{\Delta lysM}$ mice displayed no osteoclast and bone phenotypes (*Aliprantis et al., 2008*), suggesting that the deletion efficiency of *Lyz2*-Cre is weaker than *Mx1*-Cre and is allele-dependent (*Dallas et al., 2018* and *Clausen et al., 1999*). The *Ctsk*-Cre, especially the *Ctsk*-Cre knock-in line, has been reported to delete genes at the late stages of differentiated osteoclasts (*Nakamura et al., 2007*). It should be noticed that the *Ctsk*-Cre has been reported to cause gene-disruption in mesenchymal progenitor cells including periosteal cells and osteoblasts (*Yang et al., 2013* and *Ruiz et al., 2016*). Thus, the differences of skeletal phenotypes between Tfr1$^{\Delta lysM}$ and Tfr1$^{\Delta Ctsk}$ mice might be caused by the in vivo efficacy and the specificity of targeting cells of these two Cre mouse lines.

The high bone mass observed in Tfr1$^{\Delta lysM}$ and Tfr1$^{\Delta Ctsk}$ mice is more dramatic in the appendicular skeleton (femurs and tibias) (*Figure 2* and *Figure 2—figure supplement 6*) than the axial skeleton (vertebras) (*Figure 2—figure supplements 5 and 7*). The trabecular bone is more vulnerable to Tfr1-deficiency than the cortical bone. The exact explanation(s) for distinct effects of Tfr1-deficiency on different sites and compartments of bone are not available. Our qPCR result demonstrated that deletion efficiency of Tfr1 gene, *Tfrc*, in vertebra is less than femurs (*Figure 2—figure supplement 9*). It is also likely that the different physical properties, remodeling rate, microenvironment, and the heterogeneous osteoclast subtypes in the distinct sites and compartments of bone may contribute to these site- and compartment-specific phenotypes which have been reported in other genetically modified mouse models (*Cruz Grecco Teixeira et al., 2016* and *Sato et al., 2020*).

In this study, we have demonstrated that energy metabolism regulated by Tf11-mediated iron uptake is specifically indispensable for osteoclast activation and function but has little influence on osteoclast differentiation. In contrast, *Ishii et al., 2009* have reported that in vitro knockdown of Tfr1

expression in BMM by short-hairpin RNA (shRNA) inhibits osteoclast differentiation. This discrepant finding might be caused by the off-target effects of Tfr1 shRNA on expression of genes essential for osteoclast differentiation or by the distinct effects of short-term Tfr1 downregulation by shRNA vs long-term genetic deletion of Tfr1 on osteoclast lineage cells. In support of our finding, it has been recently reported that deletion of mitochondria coactivator PGC-1β in myeloid osteoclast precursor cells by *Lyz2*-Cre diminishes mitochondrial biogenesis and function in osteoclasts, leading to cytoskeletal disorganization and bone resorption retardation with normal osteoclasts differentiation (*Zhang et al., 2018*). In this study, the PGC-1β-stimulated osteoclast cytoskeleton activation is identified to be mediated by GIT1 (G protein coupled receptor kinase 2 interacting protein 1) which has been reported to regulates osteoclast function and bone mass (*Menon et al., 2010* and *Jones and Katan, 2007*). Another report by Nakano et al. has shown that deletion of G-protein Gα$_{13}$ in myeloid cells augments mitochondrial biogenesis and metabolism in osteoclasts. The gain-of-function of mitochondrial pathway induced by Gα$_{13}$-deletion in osteoclasts promotes actin cytoskeleton dynamic and bone resorption via activation of cytoskeleton regulators c-Src, Pyk2, and RhoA-Rock2. Again, Gα$_{13}$-deletion in osteoclast lineage cells has minimum effects on osteoclastogenesis in vivo and in vitro (*Nakano et al., 2019*). Intriguingly, we have unveiled that Tfr1-mediated iron uptake regulates osteoclast cytoskeleton probably via a distinct mechanism. The level of GIT1, Rac1, Cdc42, and Rock2 remains unchanged in Tfr1-null osteoclasts (*Supplementary file 5*). However, the activation of c-Src-Rac1-WRC axis by M-CSF is attenuated and mostly downregulated in the absence of Tfr1, indicating that Tfr1 modulates osteoclast cytoskeleton through promoting the stability of WRC complex. In supporting of this premise, overexpression of Hem1 (also known as Nckap1l) partially rescues the cytoskeletal defects in Tfr1-deficient osteoclasts (*Figure 9*).

In summary, we have provided evidence demonstrating that Tfr1-mediated iron uptake is a major iron acquisition pathway in osteoclast lineage cells that differentially regulates trabecular bone remodeling in perpendicular and axial bones of female and male mice. The increased intracellular iron facilitated by Tfr1 is specifically required for osteoclast mitochondrial energy metabolism and cytoskeletal organization.

# Materials and methods

## Key resources table

| Reagent type (species) or resource | Designation | Source or reference | Identifiers | Additional information |
|---|---|---|---|---|
| Strain and strain background (*Mus musculus*) | *Tfrc*-flox | Dr. Nancy C Andrews | | 129Sv |
| Strain and strain background (*Mus musculus*) | *Tfrc*-flox | Zhao Lab | | C57BL6/J |
| Strain and strain background (*Mus musculus*) | *Lyz2*-Cre | The Jackson Laboratory | cat#004781 | C57BL6/J |
| Strain and strain background (*Mus musculus*) | *Ctsk*-Cre | Dr. Takashi Nakamura | | C57BL6/J |
| Other | alpha-MEM | MilliporeSigma | M0644 | cell culture medium |
| Other | Blasticidin | MilliporeSigma | 203,350 | Antibiotics (2 µg/ml) |
| Other | DMEM | MilliporeSigma | D-5648 | cell culture medium |
| Other | Fetal bovine serum (FBS) | Hyclone | | cell culture (10%) |
| Other | 10× penicillin-streptomycin- l-glutamine | MilliporeSigma | G1146 | Antibiotics (1×) |
| Other | 10× trypsin/EDTA | Thermo-Fisher Scientific | 15400–054 | Antibiotics (1×) |
| Chemical compound and drug | Mouse apo-transferrin | MilliporeSigma | T0523 | |
| Chemical compound and drug | Dihydrotestosterone | MilliporeSigma | D-073 | $10^{-8}$ M |
| Chemical compound and drug | Estrodiol | MilliporeSigma | E2257 | $10^{-8}$ M |
| Chemical compound and drug | ferric ammonium citrate (FAC) | MilliporeSigma | F5879 | |
| Chemical compound and drug | Fe$^{59}$ ferric chloride | PerkinElmer Inc | | |

*Continued on next page*

*Continued*

| Reagent type (species) or resource | Designation | Source or reference | Identifiers | Additional information |
|---|---|---|---|---|
| Chemical compound and drug | Hemin | MilliporeSigma | H9039 | |
| Chemical compound and drug | Sodium bicarbonate | MilliporeSigma | S5761 | |
| Other | Albumin | MilliporeSigma | A7906 | Blocking reagent (0.20% in PBS) |
| Other | Alexa Fluro-488 Phalloidin | Thermo-Fisher Scientific | A12379 | Filament actin staining (1:400) |
| Other | Glycerol | MilliporeSigma | G5516 | mounting reagent (80% in PBS) |
| Other | Hoechst 33,342 | Thermo-Fisher Scientific | H3570 | Nuclei staining (1:4,000 in PBS) |
| Other | Paraformaldehyde | MilliporeSigma | P6148 | immunofluorescent staining (4% in PBS) |
| Other | Triton X-100 | MilliporeSigma | T-9284 | immunofluorescent staining (0.1%) |
| Other | NaK tartrate | MilliporeSigma | S6170 | TRAP staining |
| Other | Naphthol AS-BI (phosphoric acid solution) | MilliporeSigma | 1,802 | TRAP staining |
| Other | 3,3'-diaminobenzidine (DAB) tablets | MilliporeSigma | D-5905 | pit staining |
| Other | 30% H2O2 | MilliporeSigma | 216,763 | pit staining |
| Other | Peroxidase-conjugated WGA (wheat germ agglutinin) lectin | MilliporeSigma | L-7017 | pit staining (20 µg/ml) |
| Other | cOmplete EDTA-free protease inhibitor cocktail | MilliporeSigma | 4693159001 | protease inhibitor |
| Other | Enhanced chemiluminescent detection reagents (ECL) | MilliporeSigma | WBKLS0100 | immunoblotting |
| Other | Polyvinylidene difluoride membrane (PVDF) | MilliporeSigma | IPVH00010 | immunoblotting |
| Other | RIPA buffer | MilliporeSigma | R-0278 | cell lysate |
| Antibody | Mouse monoclonal anti-cathepsin K (clone 182–12 G5) | MilliporeSigma | MAB3324 | 1:2000 |
| Antibody | Mouse monoclonal anti-HA.11 (clone 16B12) | Biolegend | 901,513 | 1:5000 |
| Antibody | Mouse monoclonal anti-Tfr1 (clone H68.4) | Thermo-Fisher Scientific | 13–6800 | 1:500 |
| Antibody | Mouse monoclonal anti-tubulin (clone DM1A) | MilliporeSigma | T9026 | 1:3000 |
| Antibody | Mouse monoclonal Total OXPHOS Antibody Cocktail | Abcam | ab110413 | 1:1000 |
| Antibody | Goat polyclonal HRP-anti-mouse secondary antibody | Cell Signaling Technology | 7,076 | 1:5000 |
| Antibody | Goat polyclonal HRP-anti-rabbit secondary antibody | Cell Signaling Technology | 7,074 | 1:5000 |
| Commercial assay or kit | High-capacity cDNA reverse transcription kit | Thermo-Fisher Scientific | 4368813 | |
| Commercial assay or kit | RNeasy mini kit | Qiagen | 74,104 | |
| Commercial assay or kit | Serum TRAcP-5b | Immunodiagnostic Systems | SB-TR103 | |
| Commercial assay or kit | RatLaps (CTx-I) EIA | Immunodiagnostic Systems | AC-06F1 | |
| Commercial assay or kit | Rat/mouse PINP EIA | Immunodiagnostic Systems | AC-33F1 | |
| Commercial assay or kit | Hepcidin Murine-Complete ELISA Kit | Intrinsic Lifesciences | HMC-001 | |
| Commercial assay or kit | Pointe Scientific Iron/TIBC Reagents | Fisherscientific | 23-666-320 | |
| Commercial assay or kit | Iron Assay Kit | Abcam | ab83366 | |
| Commercial assay or kit | Active Arf1 pull-down and detection kit | Thermo-Fisher Scientific | 16,121 | |

*Continued on next page*

*Continued*

| Reagent type (species) or resource | Designation | Source or reference | Identifiers | Additional information |
|---|---|---|---|---|
| Commercial assay or kit | Active Rac1 pull-down and detection kit | Thermo-Fisher Scientific | 16,118 | |
| Commercial assay or kit | TMTsixplex isobaric Mass Tagging kit | Thermo-Fisher Scientific | 90,064 | |
| Other | MitoTracker Green fluorescence | Thermo-Fisher Scientific | M7514 | Mitochondrial assay reagent |
| Other | MitoSOX Red | Thermo-Fisher Scientific | M36008 | Mitochondrial assay reagent |
| Other | JC-1 Dye | Thermo-Fisher Scientific | T3168 | Mitochondrial assay reagent |
| Recombinant DNA reagent | Murine *Nckap1l* | Dharmacon Inc | MMM1013-202769283 | cDNA template |
| Commercial assay or kit | TransIT-LT1 | Mirus Bio LLC | MIR2300 | DNA transfection reagent |
| Commercial assay or kit | *Acp5* | Thermo-Fisher Scientific | Mm00475698_m1 | Quantitative PCR (qPCR) primer |
| Commercial assay or kit | *Ctsk* | Thermo-Fisher Scientific | Mm00484039_m1 | qPCR primer |
| Commercial assay or kit | *mt-Cox1* | Thermo-Fisher Scientific | Mm00432648_m1 | qPCR primer |
| Commercial assay or kit | *Dcstamp* | Thermo-Fisher Scientific | Mm04209236_m1 | qPCR primer |
| Commercial assay or kit | *Nfatc1* | Thermo-Fisher Scientific | Mm00479445_m1 | qPCR primer |
| Commercial assay or kit | *Mfsd7c* | Thermo-Fisher Scientific | Mm01302920_m1 | qPCR primer |
| Commercial assay or kit | *Mrps2* | Thermo-Fisher Scientific | Mm00475529_m1 | qPCR primer |
| Commercial assay or kit | *Runx2* | Thermo-Fisher Scientific | Mm00501584_m1 | qPCR primer |
| Commercial assay or kit | *Slc11a2* | Thermo-Fisher Scientific | Mm00435363_m1 | qPCR primer |
| Commercial assay or kit | *Slc39a14* | Thermo-Fisher Scientific | Mm01317439_m1 | qPCR primer |
| Commercial assay or kit | *Slc40a1* | Thermo-Fisher Scientific | Mm01254822_m1 | qPCR primer |
| Commercial assay or kit | *Slc46a1* | Thermo-Fisher Scientific | Mm00546630_m1 | qPCR primer |
| Commercial assay or kit | *Tfrc* | Thermo-Fisher Scientific | Mm00441941_m1 | qPCR primer |
| Commercial assay or kit | *Tfr2* | Thermo-Fisher Scientific | Mm00443703_m1 | qPCR primer |

The animal work follows the ARRIVE guidelines. All the in vivo and in vitro experiments were performed and analyzed in double-blinded manner.

## Mice and genotyping

*Tfrc*-flox congenic mice on a 129Sv background were kindly provided by Dr. Nancy C Andrews (Duke University). *Tfrc*-flox mice on a homogeneous C57BL/6 background were generated by backcrossing 129Sv *Tfr1*-flox mice with C57BL/6 mice for more than 10 generations. *Lyz2*-Cre mice on a C57BL/6 background (B6.129P2tm1/cre, stock number 004781) were purchased from The Jackson Laboratory (Bar Harbor, ME, USA). *Ctsk*-Cre congenic mice on a C57BL/6 background were obtained from Dr. Takashi Nakamura (Keio University, Tokyo, Japan). The primers and PCR protocols for genotyping *Tfrc*-flox and *Lyz2*-Cre mice followed those provided by The Jackson Laboratory. The following primers for genotyping *Ctsk*-Cre mice were used: P1N (5'-CCTAATTATTCCTTCCGCCAGGATG-3'), P2N (5'-CCAGGTTATGGGCAGAGATTTGCTT-3'), and P3N (5'-CACCGGCATCAACGTTTTCTTTTCG-3'). In vivo analyses of skeletal phenotypes were performed on F2 mice of mixed and homogeneous backgrounds.

## Reagents and antibodies

See Key resources table.

## Micro computed tomography

The left femurs, tibias, and L4 vertebrae were cleaned of soft tissues and fixed in 4% paraformaldehyde in PBS for overnight at 4°C. After washing with PBS for three times, the bones were stored in PBS with 0.02% sodium azide. The bones were loaded into a 12.3 mm diameter scanning tube and were imaged in a µCT (model µCT40, Scanco Medical). We integrated the scans into 3-D voxel images (1024×1024 pixel matrices for each individual planar stack) and used a Gaussian filter (sigma = 0.8 and

support = 1) to reduce signal noise. A threshold of 200 was applied to all scans, at medium resolution (E = 55 kVp, I = 145 µA, and integration time = 200 ms).

## Histology and bone histomorphometry

The left femurs were embedded undecalcified in methyl methacrylate. The dynamic histomorphometric examination of trabecular bone formation was done on 5 µm longitudinal sections with a digitizer tablet (OsteoMetrics, Inc, Decatur, GA, USA) interfaced to a Zeiss Axioscope (Carl Zeiss, Thornwood, NY, USA) with a drawing tube attachment. The right femurs fixed and decalcified in 14% EDTA for 7–10 days. The bones were embedded in paraffin before obtaining 5 µm longitudinal sections. After removal of paraffin and rehydration, sections were stained for TRAP activity and counter-stained with fast green, and osteoclasts were enumerated on the trabecular bone surface using Osteomeasure histomorphometric software (OsteoMetrics, Inc, Decatur, GA, USA).

## Serum TRAcP-5b, CTx-I, and PINP ELISA

Blood was collected retro-orbitally under inhalation of a 2% isoflurane/oxygen mix anesthesia immediately prior to sacrifice. Serum was obtained by centrifugation of blood in a MiniCollect tube (catalog no. 450472, Greiner Bio-one GmbH, Austria). The serum levels of TRAcP-5b, CTx-I, and PINP were measured by a mouse TRAP (TRAcP 5b) kit (SB-TR103), RatLaps (CTx-I) EIA (AC-06F1), and rat/mouse PINP EIA kit (AC-33F1) from Immunodiagnostic Systems following their instructions.

## Serum hepcidin ELISA and serum total iron measurement

The serum level of hepcidin in control and Tfr1$^{\Delta LysM}$ mice was assayed by the Hepcidin Murine-Complete ELISA Kit from Intrinsic Lifesciences (La Jolla, CA, USA). The serum total iron concentration was measured using a Poine Scientific Iron kit.

## Ovariectomy

The 4.5-month-old control and Tfr1$^{\Delta LysM}$ female mice were anesthetized by a 0.5% isoflurane/oxygen mix and underwent bilateral OVX using the dorsal approach. The sham-operated mice had the ovaries exteriorized but not removed. Uterine weight was determined at sacrifice to verify the successful removal of the ovaries. 6 weeks after operation, mice were sacrificed, and femurs were isolated and fixed by 4% paraformaldehyde/PBS overnight before scanned by µCT.

## In vitro osteoclast cultures

Whole bone marrow was extracted from tibia and femurs of 6- to 8-week-old control and conditional knockout male and female mice. Red blood cells were lysed in buffer (150 mM NH4Cl, 10 mM KNCO3, 0.1 mM EDTA, and pH 7.4) for 5 min at room temperature. $5\times10^6$ bone marrow cells were plated onto a 100 mm petri-dish and cultured in α–10 medium (α-MEM, 10% heat-inactivated fetal bovine serum [FBS], 1×PSG, penicillin-streptomycin-glutamine) containing 1/10 volume of CMG 14–12 (conditioned medium supernatant containing recombinant M-CSF at 1 µg/ml) for 4–5 days. BMM were lifted by 1×trypsin/EDTA and replated at density of 160/mm$^2$ onto tissue culture plates or dishes with 1/100 vol of CMG 14–12 culture supernatant along for monocytes or cultured with 1/100 vol of CMG 14–12 culture supernatant plus 100 ng/ml of recombinant RANKL for 2 and 4 days to generate mononuclear pre-osteoclasts and mature osteoclasts, respectively.

## Retroviral transduction

A total of 3 µg of pMX (mammalian expression) retroviral vector or recombinant pMX vector expressing HA-tagged murine Hem1 was transfected into Plat E retroviral packing cells using TransIT-LT1 transfection reagent. Virus supernatants were collected at 48 hr after transfection. $5\times10^6$ bone marrow cells isolated from Tfr1 myeloid conditional knockout mice were plated onto a 100 mm petri-dish and cultured in α–10 medium (α-MEM, 10% heat-inactivated FBS, 1×PSG) containing 1/10 volume of CMG 14–12 supernatant for two days. BMM were transduced with viruses for 24 hr in α–10 medium containing 1/10 volume of CMG 14–12 supernatant and 20 µg/ml of protamine. Cells were then lifted by 1×trypsin/EDTA and replated at $2\times10^6$ density onto a 100 mm petri-dish. The positively transduced cells were selected in α–10 medium containing M-CSF and 1.5 µg/ml of blasticidin for 3 days.

## TRAP staining

Osteoclasts cultured on 48-well tissue culture plate were fixed with 4% paraformaldehyde/PBS for 20 min at room temperature. After washing with PBS for 5 min twice, TRAP was stained with NaK tartrate and naphthol AS-BI phosphoric acid. Photomicrographs were taken with a stereomicroscope with a digital camera (Discovery V12 and AxioCam; Carl Zeiss, Inc). The number of osteoclasts with more than three nuclei was counted and analyzed by GraphPad Prism 6 in a double-blinded manner.

## Resorption pit staining

Mature osteoclasts grown on cortical bovine bone slices were fixed with 4% paraformaldehyde/PBS for 20 min. After washing in PBS for 5 min twice, cells were removed from bone slices with a soft brush. The slices were then incubated with 20 µg/ml peroxidase-conjugated wheat germ agglutinin lectin for 60 min at room temperature. After washing in PBS twice, bone chips were incubated with 0.52 mg/ml 3,3_-diaminobenzidine and 0.03% H2O2 for 30 min. Samples were mounted with 80% glycerol/PBS and photographed with Zeiss AxioPlan2 microscope equipped with an Olympus DP73 digital camera. The resorbed area/bone slice was quantified by ImageJ software (National Institutes of Health), and the percentage of pit area vs that of the whole bone slice was calculated and analyzed by GraphPad Prism 6.

## Fluorescent staining of actin filaments and nuclei

Osteoclasts cultured on glass coverslips or bone slices were fixed with 4% paraformaldehyde in PBS for 20 min and permeabilized with 0.1% Triton X-100/PBS for 10 min at room temperature. The filament actin and nuclei were labeled by Alexa-488 conjugated Phalloidin (1:100 from a 1 mg/ml stock) and Hoechst 33,342 (1:4000 from a 10 mg/ml stock), respectively, for 15 min at room temperature. After two times 5-min wash with PBS, samples were mounted with 80% glycerol/PBS and photographed under a Zeiss AxioImager Z1 fluorescent microscope equipped with Zeiss AxioCam MRm monochromatic and MR5c color cameras with a set of fluorescent filters. The percentage of active osteoclasts (podosome-belt bearing osteoclasts on glass coverslips and actin-ring bearing osteoclasts on bone slices) out of total osteoclasts was calculated and analyzed by GraphPad Prism 6.

## RNA isolation and real-time qPCR

Total RNA was purified using RNeasy mini kit (Qiagen) according to the manufacture's protocol. First-strand cDNAs were synthesized from 0.5 to 1 µg of total RNA using the High-Capacity cDNA Reverse Transcription kits (Thermo-Fisher Scientific) following the manufacturer's instructions. TaqMan real-time qPCR was performed using the primers from Thermo-Fisher Scientific (Key resources table). Samples were amplified using the StepOnePlus real-time PCR system (Life Technologies) with an initial denaturation at 95°C for 10 min, followed by 40 cycles of 95°C for 15 s and 60°C for 1 min. The relative cDNA amount was calculated by normalizing to that of the mitochondrial gene Mrps2, which is steadily expressed in both BMM and osteoclasts, using the ΔCt method.

## Immunoblotting

Cells were washed with ice-cold PBS twice and lysed in 1× RIPA buffer containing cOmplete Mini EDTA-free protease inhibitor cocktail. After incubation on ice for 30 min, cell lysates were clarified by centrifugation at 14,000 rpm for 15 min at 4°C. 10–30 µg of total protein was subjected to 8 or 10% SDS-PAGE gels and transferred electrophoretically onto polyvinylidene difluoride membrane by a semi-dry blotting system (Bio-Rad). The membrane was blocked in 5% fat-free milk/Tris-buffered saline for 1 hr and incubated with primary antibodies at 4°C overnight followed by horseradish peroxidase conjugated secondary antibodies. After rinsing three times with Tris-buffered saline containing 0.1% Tween 20, the membrane was incubated with enhanced chemiluminescent for 5 min.

## $^{59}$Fe-transferrin uptake and colorimetric total cellular iron measurement

Mouse apo-transferrin was labeled with $^{59}$Fe and gel-filtered on a Sephadex G-50 column. For Tf-dependent $^{59}$Fe influx measurements, 25 µg of $^{59}$Fe-labeled Tf was added to each well of a 6-well plate with 3 ml of medium and incubated in a CO2 incubator (37°C, 5% CO2) with rotation at 80 rpm. At various times after adding labeled Tf, the medium was aspirated, and cells were washed gently three

times with 2 ml of cold PBS. Wells were extracted with 1 ml of 0.1 N NaOH; radioactivity was determined on a 0.5 ml aliquot by a gamma counter. Protein was determined on 25 µl of the extract.

To measure cellular iron, cells cultured in 6-well plates were collected in 50 µl of the Iron Assay Buffer provided in an Iron Assay kit (Abcam) and were homogenized with pellet pestles associated with a motor. The cells lysates were centrifuged at 15,000 rpm for 10 min. The supernatants proceeded to measure the total iron ($Fe^{2+}$ and $Fe^{3+}$) concentration following the manufacturer's protocol.

## Quantitative proteomic and mass-spectrometry

Cells were harvested and lysed in 2% SDS lysis buffer. A total of 100 µg of proteins in each cell lysate was reduced, alkylated, and digested using filter-aided sample preparation. Tryptic peptides were cleaned by solid phase extraction (SPE), normalized and labeled using a TMTsixplex isobaric Mass Tagging kit. The labeled peptides were cleaned by SPE and mixed. The mixed peptides from each cell culture were separated into 36 fractions on a 100 × 1.0 mm Acquity BEH C18 column (Waters) using an UltiMate 3000 UHPLC system (Thermo-Fisher Scientific) with a 40 min gradient from 99:1 to 60:40 buffer A:B (Buffer A contains 0.5% acetonitrile and 10 mM ammonium hydroxide). Buffer B contains 10 mM ammonium hydroxide in acetonitrile ratio under basic (pH 10) conditions and then consolidated into 13 super-fractions.

Each super-fraction was then further separated by reverse phase XSelect CSH C18 2.5 mm resin (Waters) on an in-line 150 × 0.075 mm column using an UltiMate 3000 RSLCnano system (Thermo). Peptides were eluted using a 60 min gradient from 97:3 to 60:40 buffer A:B ratio. Here, buffer A contains 0.1% formic acid and 0.5% acetonitrile, and buffer B contains 0.1% formic acid and 99.9% acetonitrile. Eluted peptides were ionized by electrospray (2.15 kV) followed by mass spectrometric analysis on an Orbitrap Fusion Lumos mass spectrometer (Thermo) using multi-notch MS3 parameters. The mass spectrometry data were acquired using the FTMS (fourier transform mass spectrometer) analyzer in top-speed profile mode at a resolution of 120,000 over a range of 375–1500 m/z. Following CID activation with normalized collision energy of 35.0, MS/MS data were acquired using the ion trap analyzer in centroid mode and normal mass range. Using synchronous precursor selection, up to 10 MS/MS precursors were selected for HCD activation with normalized collision energy of 65.0, followed by acquisition of MS3 reporter ion data using the FTMS analyzer in profile mode at a resolution of 50,000 over a range of 100–500 m/z.

Proteins were identified and reporter ions quantified by searching the UniprotKB Mouse database using MaxQuant (version 1.6.10.43, Max Planck Institute) with a parent ion tolerance of 3 ppm, a fragment ion tolerance of 0.5 Da, a reporter ion tolerance of 0.001 Da, trypsin/P enzyme with two missed cleavages, variable modifications including oxidation on M and acetyl on protein N-term, and fixed modification of carbamidomethyl on C. Protein identifications were accepted if they could be established with less than 1.0% false discovery. Proteins identified only by modified peptides were removed. Protein probabilities were assigned by the Protein Prophet algorithm. TMT MS3 reporter ion intensity values were analyzed for changes in total protein.

## Mitochondrial mass, ROS production, and membrane potential measurements

Mitochondrial content, mitochondria-derived ROS, and mitochondrial membrane potential were measured using MitoTracker Green fluorescence, MitoSOX Red Mitochondrial Superoxide Indicator, and JC-1 Dye from Thermo-Fisher Scientific, respectively, using fluorescence microscopy. Briefly, cells cultured onto glass-bottom imaging dishes were incubated with MitoTracker Green (50 nM), MitoSOX (5 µM), or JC-1 (5 µg/ml), respectively, for 15 min at 37°C. The probes were then washed off, and cells were examined under fluorescent microscope. The epifluorescent images of five randomly chosen areas per group (control or knockout) were taken in a blinded fashion. Each area had between 5 and 10 cells. The images were analyzed with Image J software also in a blinded fashion. The fluorescent intensities of cells were individually measured, and the results were presented as mean fluorescent intensity per cell in arbitrary units.

## Seahorse mitochondrial flux analysis

BMM were plated in wells of Seahorse XF96 cell culture plates. The cells were cultured with 10 ng/ml M-CSF alone or 10 ng/ml M-CSF plus 100 ng/ml recombinant RANKL for 2 days to generate

monocytes and pre-osteoclasts or 4 days for mature osteoclasts. On the day of respiratory function measurements, the media in the wells were changed to unbuffered Dulbecco's modified Eagle's medium supplemented with 4 mM glutamate and incubated in a non-CO2 incubator for 1 hr at 37°C. Three baseline measurements were acquired before injection of mitochondrial inhibitors or uncouplers. OCR measurements were taken after sequential addition of oligomycin (10 µM), FCCP (5 µM), and rotenone/antimycin A mixture (10 µM). OCRs were calculated by the Seahorse XF-96 software and represent an average of three measurements on 18–24 different wells. The rate of measured oxygen consumption was reported as fmol of O2 consumed per min per cell.

## Rac1 and Arf1 activation assay

BMM obtained from the control and Tfr1$^{\Delta LysM}$ mice were cultured with M-CSF and RANKL for 3 days to generate pre-osteoclasts. The cells were then under serum and cytokine starvation for 6 hr before stimulated with 50 ng/ml M-CSF and 100 ng/ml RANKL, respectively. The activation of small GTPases Rac1 and Arf1 was assayed by GST-pull down assay kits from Thermo-Fisher Scientific Inc.

## Statistics

Based on power analysis using SD and the population variance estimated from our previous studies and reports by others, the required sample size of mice for $\alpha=0.05$ (two-sided) and a power of 0.95 is ≥6 animals per group. The in vivo biological replicates are defined as an individual mouse for each experiment. All in vitro data were representatives of individual biological replicates from independent experiments and not technical replicates (repeated measurements of the same sample). For all graphs, data are represented as the mean ± SD. For comparison of two groups, data were analyzed using a two-tailed Student's t-test. For comparison of more than two groups, data were analyzed using one-way ANOVA, and the Bonferroni procedure was used for Tukey comparison. For all statistical tests, the analysis was performed using Prism 6 (GraphPad Software, La Jolla, CA), and a p value of less than 0.05 was considered significant.

## Study approval

All animal protocols and procedures used in animal studies were approved by the Institutional Animal Care and Use Committees of the University of Arkansas for Medical Sciences, Long Beach VA Healthcare System, and Loma Linda VA Healthcare System (IACUC #1,685 and #1774). The protocols for generation and use of recombinant DNAs and retroviruses were approved by Institutional Biosafety Committee of Long Beach VA Healthcare System (approval #1774).

## Acknowledgements

We would like to thank Dr Nancy C Andrews and Dr Takashi Nakamura for kindly providing *Tfr1*-flox mice and *Ctsk*-Cre mice, respectively. Erin Hogan at the University of Arkansas for Medical Sciences and Neil T Hoa at the Long Beach VA Healthcare System are acknowledged for help with microscopes. We thank Nancy Lowen and Sheila Pourteymoor for technical assistance. NAB and KJK were supported by NIH grant P20 GM 109005. The University of Arkansas for Medical Sciences Proteomics Core is supported by NIH grants R24GM137786 and P20 GM121293. LG was supported by the Research Scholar Grant funded by the American Cancer Society and the VA Merit Award BX004841. SM is a recipient of a Senior Research Career Scientist Award from the Department of Veterans Affairs and has grant support from NIH/NIAMS R01AR048139, AR070806 and VA merit BX005262. The work was supported by grants from NIH/NIAMS R01AR062012, R21AR068509, R01AR073298 to HZ.

## Additional information

### Competing interests

Subburaman Mohan: Reviewing editor, *eLife*. The other authors declare that no competing interests exist.

## Funding

| Funder | Grant reference number | Author |
| --- | --- | --- |
| American Cancer Society | RSG-17-069-01-TBG | Ling Gao |
| National Institutes of Health | R01AR073298 | Haibo Zhao |
| National Institutes of Health | R01AR078843 | Weirong Xing |
| National Institutes of Health | R01AR048139 | Subburaman Mohan |
| National Institutes of Health | R01AR070806 | Subburaman Mohan |
| National Institutes of Health | P20 GM 109005 | Nukhet Aykin-Burns |
| National Institutes of Health | R21AR068509 | Haibo Zhao |
| National Institutes of Health | R01AR062012 | Haibo Zhao |
| US Department of Veterans Affairs | VA merit BX005262 | Subburaman Mohan |
| US Department of Veterans Affairs | VA Merit Award BX004841 | Ling Gao |

The funders had no role in study design, data collection and interpretation, or the decision to submit the work for publication.

## Author contributions

Bhaba K Das, Lei Wang, Toshifumi Fujiwara, Conceptualization, Data curation, Formal analysis, Investigation, Methodology, Validation, Writing - original draft; Jian Zhou, Tomasa Barrientos, Data curation, Formal analysis, Investigation, Validation; Nukhet Aykin-Burns, Renny Lan, Data curation, Formal analysis, Investigation, Methodology, Visualization; Kimberly J Krager, Jyoti Gogoi, Data curation, Formal analysis, Investigation; Samuel G Mackintosh, Data curation, Formal analysis, Investigation, Methodology, Validation; Ricky Edmondson, Data curation, Formal analysis, Investigation, Methodology; Michael L Jennings, Data curation, Formal analysis, Investigation, Methodology, Writing - review and editing; Xiaofang Wang, Data curation, Formal analysis, Investigation, Visualization; Jian Q Feng, Data curation, Formal analysis, Investigation, Validation, Writing - review and editing; Aarthi Kannan, Data curation, Formal analysis, Investigation, Project administration; Ling Gao, Conceptualization, Data curation, Formal analysis, Investigation, Validation, Writing - review and editing; Weirong Xing, Data curation, Formal analysis, Investigation, Methodology, Validation, Writing - review and editing; Subburaman Mohan, Conceptualization, Data curation, Formal analysis, Investigation, Methodology, Validation, Writing - original draft, Writing - review and editing; Haibo Zhao, Conceptualization, Data curation, Formal analysis, Funding acquisition, Investigation, Methodology, Project administration, Supervision, Validation, Visualization, Writing - original draft, Writing - review and editing

## Author ORCIDs

Bhaba K Das http://orcid.org/0000-0003-0256-5489
Nukhet Aykin-Burns http://orcid.org/0000-0001-8574-4102
Subburaman Mohan http://orcid.org/0000-0003-0063-986X
Haibo Zhao http://orcid.org/0000-0003-0836-7555

## Ethics

All animal protocols and procedures used in animal studies were approved by the Institutional Animal Care and Use Committees of the University of Arkansas for Medical Sciences, Long Beach VA Healthcare System, and Loma Linda VA Healthcare System (IACUC #1685 and #1774). The protocols for generation and use of recombinant DNAs and retroviruses were approved by Institutional Biosafety Committee of Long Beach VA Healthcare System. 832 (approval #1774).

Decision letter and Author response
Decision letter https://doi.org/10.7554/eLife.73539.sa1
Author response https://doi.org/10.7554/eLife.73539.sa2

## Additional files

### Supplementary files
• Supplementary file 1. Bone marrow monocytes qProteomics.

• Supplementary file 2. Pre-osteoclasts qProteomics.

• Supplementary file 3. Osteoclasts qProteomics.

• Supplementary file 4. Ingenuity pathway analysis (IPA) of canonical pathways in osteoclasts.

• Supplementary file 5. Changes of cytoskeletal proteins in TfR1-null osteoclast lineage cells revealed by qProteomics.

• MDAR checklist

### Data availability
All data generated or analysed during this study are included in the manuscript and supporting file.

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
