## [Editor Report]

The current work addresses an important unknown in bone homeostasis, aiming to understand the mechanism of iron-mediated effects on bone, and presents findings of novel significance that are of interest to basic iron biologists, bone biologists, experts in mitochondrial respiration, and endocrinologists. The authors specifically focus on the role of iron deficiency via TFR1 loss in osteoclastogenesis and osteoclast function, demonstrating that cell-selective loss of TFR1 in mature osteoclasts (using Ctsk-Cre mice) leads to decreased osteoclast function and increased bone volume in long bones, especially in female mice, without influencing osteoclastogenesis. In vitro studies confirm an actin cytoskeletal defect and defect in bone resorption in TFR1 null osteoclasts, both partially rescued by heme but not NTBI. Together, these findings contribute to the knowledge in the field, supporting a hypothesis that iron delivered via TF-TFR1 (or heme iron) is essential for osteoclast function.

---

## [Decision Letter]

**Decision letter after peer review:**

Thank you for submitting your article "Transferrin receptor 1-mediated iron uptake regulates bone mass in mice via osteoclast mitochondria and cytoskeleton" for consideration by *eLife*. Your article has been reviewed by 3 peer reviewers, one of whom is a member of our Board of Reviewing Editors, and the evaluation has been overseen Mone Zaidi as the Senior Editor. The following individual involved in review of your submission has agreed to reveal their identity: Nathan J Pavlos (Reviewer #2).

Essential revisions:

The reviewers found significant merit in the work. However, the evaluation identified sufficient weaknesses that prevent publication of this work in the present form. The main shortcomings of the presented work include the relatively underdeveloped in vitro data which does not convincingly support the conclusions; the concern that the mouse models result in decreased Ctsk expression which may along at least in part explain the phenotype; the potential of evaluation in an ovarectomized mice and / or mice at different ages to extend the translational potential and underlying mechanism of these findings, respectively; and some degree of evaluation of systemic iron metabolism and erythropoiesis. Specifically, the in vitro data can be extended to more fully explore osteoclastogenesis and in vivo data by experimentally addressing the observed gender disparity and specificity of effect in the trabecular region of long bones.

*Reviewer #1 (Recommendations for the authors):*

Some additional questions and clarifications are delineated below.

1. It is unclear why only or mostly female osteoclast-selective TFR1-null mice exhibit a more robust phenotype, with increased bone mass in long bones. The authors discuss this point in the Discussion section but the proposed experiment, to evaluate whether hepcidin expression is altered in the liver or hepcidin concentration changes in the serum and IHC using anti-FPN1 antibody in osteoclasts or at least cellular iron concentration would be useful to corroborate the hypothesis of estrogen-mediated changes in the hepcidin:ferroportin axis. Do osteoclasts even express FPN1 to make such a hypothesis viable? There is some data on testosterone and hepcidin regulation (Latour Hepatology 2014) that may be worth mentioning or discussing in the setting of the authors hypothesis about differential gender-related results. Finally, are the differences between male and female mice significant in Figure 2? Are these differences in the expected range for male and female mice?

2. It is similarly unclear why only long bones but not vertebrae reveal these differences. Can the authors speculate and discuss this point in the Discussion section?

3. The authors suggest that they have effectively excluded an effect from TFR2 on osteoclast function. However, this is inaccurate and needs to be communicated differently. At best, the authors have confirmed what has long been understood that TFR2 serves as an iron sensor rather than importer of iron. This should be restated in all mention of TFR2 in the manuscript. Furthermore, Rauner et al. published an important manuscript looking at the role of TFR2 in bone, demonstrating that TFR2 competes with BMPR for binding BMP2 and TFR2:BMP binding is enhanced in the presence of Fe-TF, together suggesting that presence of iron may favor bone resorption in the presence of TFR2. Something akin to this would benefit the readership and should be included in the manuscript as it pertains to the data on TFR1 here.

4. Figure 1, Y axis on mRNA expression panels is unclear what the control is. Typically, expectation is of change from baseline (1) in control. Minor: panel for Flvcr2 is missing from legend.

5. Figure 1 B, what mechanism leads to some cellular iron in TFR1 fl/LysM Cre OC cells? This may be the reason that the results on differentiation are not seen in the current work relative to Ishii et al. 2009. Please explain and include in the Discussion.

6. Figure 3 A-E is redundant and can be moved to the Supplement. In general, the data on LysM-cre mice is distracting and does not provide substantive contributing information to the main point. Consider moving all LysM-cre data to the supplement to better focus the manuscript.

7. Figure 3G, the difference in female mice is driven by 3 mice with elevated values; are these 3 mice in any way different from the others? Are they outliers?

8. Figure 3 states that there is no impact on osteoclast number and bone resorption in mice but while BFR is unchanged in Figure 3J, serum P1NP is increased. How do the authors reconcile this confusing result?

9. Given the number of figures, it would be worthwhile to consolidate and reorganize to move some additional elements into the supplementary figures. For example, there is no reason in the mind of the reviewer to include both male and female samples in vitro. Especially since there are no visible differences, please move one gender to the supplement.

10. Figure 6 title suggests that there are no differences in markers of differentiation, e.g. Nfatc1 and Ctsk, but both of these appear to be decreased in f/f;c/c samples, both female but especially male (especially since tubulin appears greater in those samples). Please provide the quantification here.

11. Figue 12B and 12C do not include control cells with and without Hem1 overexpression. This is important since the mechanism may be completely different and yet additive to yield the observed results. Please edit.

12. The Introduction is too long and unfocused. Please edit.

13. The first half of the Discussion is a restating of the results and needs to be re-written to further extend the interpretation of the results without restating them.

*Reviewer #2 (Recommendations for the authors):*

This is a well-written study that examines the role of transferrin receptor 1 in osteoclast function and bone remodelling. The importance of iron homeostasis in bone turnover has gained momentum in recent years, most notably the role of transferrin receptor 2 in osteoblasts (Rauner et al., Nat Med 2019), however the physiological significance of iron and the mechanism(s) responsible for its uptake in osteoclasts remains poorly understood. As such, the present study detailing the physiological role of Tfr1 in osteoclasts would likely be of considerable interest to the field, complementing existing studies in osteoblasts, but also reconciling the physiological contribution of cellular iron handling on the other side of the bone remodelling equation. The strengths of this study lie the multiple osteoclast-specific mouse lines generated to interrogate the role of Tfr1 in both osteoclast-lineage cells and mature osteoclasts along with the suite of complementary cell biological, biochemical and proteomic analyses used to dissect its contribution in osteoclasts. Nonetheless, there remain some weaknesses, particularly concerning the expressivity of the bone phenotype and mechanistic link between Tfr1 and the cytoskeleton that should be considered to strengthen the overall resolve of the conclusions drawn.

I have the following comments:

1. The studies conducted on LysoM-Cre mice appear to be well-executed and carefully controlled. The skeletal phenotype is however complex, being restricted to trabeculae of the long bones of female mice, but not the axial skeleton (e.g. vertebrae) at 10-weeks of age. The authors rationalise that the unexpected sexual dimorphism is associated with an intersection between estrogen and iron export, which is reasonable, but they do not offer experimental evidence to support this. Further, the skeletal phenotyping is limited to a single time-point i.e. 10-weeks of age. As such, it would be beneficial if the authors could provide data in older animals to show whether the bone phenotype becomes more pronounced or resolves with age. It would also be useful for the reader if the authors could offer some explanation for the observed site-specific differences in the Discussion e.g. potential differences in osteoclast heterogeneity etc.

2. Although widely employed as a Cre-transgenic mouse line to conditionally delete target genes in mature osteoclasts there are some caveats that should be considered to avoid over-interpretation of data arising from the cathespin K (Ctsk)-Cre lines. Foremost, this cre-line is known to result in the loss one allele of Ctsk owing to partial replacement the Ctsk locus with Cre in the targeting cassette (Dallas et a., Genetics 2018). Thus, endogenous Ctsk expression is reduced. Considering that deficiency and mutations in Ctsk lead to high bone mass phenotype (Saftig et al., 1997) together with findings that Cstk K-Cre can also lead to unexpected germline deletion of genes (e.g. Winkeler et al., PLos 2012), any data arising from these mice need to be interpreted with a degree of caution. The authors have rightly attempted to control for this using littermates mice on heterozygous KI (Ctsk-Cre/+), which is appropriate although these mice are on a mice mixed background C57BL6/129 which can complicate skeletal phenotypes. Overall, these underlying complexities may account, in part, for the discrepancies observed between the LysoM and Ctsk-cre males and females, with Ctsk-Cre leading to a more pronounced phenotype. As a minimum, these limitations need to be acknowledged and discussed. A minor oversight- for Figure 4B is that the mice sexes are unlabelled in the reconstructed uCT scans.

3. The in vitro data concerning the potential effects of Tfr1-deficiency on osteoclast different and mitochondrial metabolism are extensive and overall support the conclusion drawn. On the other hand the mechanism linking Tfr1 regulation of the osteoclast cytoskeleton remains largely descriptive and thus is comparatively underdeveloped. While the quantitative proteomics data goes some way to lend support to the notion that loss of Tfr1 leads to a global dysregulation of downstream Tfr1-associated protein pathways including those involved with cytoskeletal signalling/organization, the extent and diversity of proteins affected in osteoclasts >2000 proteins is remarkable. Can the authors please clarify how they arrived at the 1.2-fold threshold for these data. Does the 1.2-fold represent an arbitrary cut-off or does it reflect >1.2 Log2Fold, p<0.05? To validate these proteomic data the authors hone in on hem1 (a component of the Wave regulatory complex, WRC) and show that it partly restores the cytoskeletal spreading defect in Tfr1-deficient osteoclasts. This finding is interesting but would be bolstered by confirmation of a disruption of additional components of the WRC or downstream components such as Arf2/3 which has an established role in OC cytoskeletal organization in osteoclasts (Hurst et al., JBMR 2009). Considering the well-established roles of B3-integrin, c-SRC and Pyk2 in osteoclasts, it would be valuable to corroborate/or discount their direct involvement in these spreading and cytoskeletal defects by independently assessing their expression levels immunoanalytically in Tfr1-deficient osteoclasts.

*Reviewer #3 (Recommendations for the authors):*

1. The authors state that Tfr1 does not inhibit osteoclast differentiation, but just spreading and resorption. I do not agree with this statement. By "spreading", do the authors mean "fusion"? I believe in any case that this is part of the differentiation process to a fully mature osteoclast. Also, even though it is stated that osteoclasts have been counted as TRAP-positive and with more than 3 nuclei, I know from experience that it is tricky to identify the nuclei just using TRAP staining, especially in such small cells as presented in Suppl. Figure 9A. I believe it may be more accurate to assess the number of osteoclasts in this case with so many small ones using IF and DAPI staining for the nuclei. Using that, the authors could also count how many osteoclasts have e.g. 3-10 or 10-50 nuclei (giving insights into the fusion process). Finally, I believe more markers for osteoclasts could be used to address this point of fusion/spreading, such as DC-STAMP, SWAP70, or CTSK. In Suppl. Figure 9, e.g., it looks like CtsK expression is reduced in the Tfr1;Lysm-cre mice, although the WB is a bit over-exposed, so it is hard to tell… quantification would be helpful.

2. It appears that the appendicular skeleton is more affected than the axial skeleton. Did the authors check the expression of Tfr1 in e.g. femoral vs. vertebral bone using immunohistochemistry? Is there a difference in expression pattern that may explain the altered susceptibility of the sites to Tfr1 deletion? Similarly, it would be interesting to check Tfr1 in trabecular vs. cortical bone – as those also seem to be differentially affected.

3. Along those lines, is the expression of Tfr1 in osteoclasts (bone) lower in young animals compared to adult ones? Could that explain the more severe effects of iron deficiency in adult mice compared to the 3-week-old ones?

4. The increase in BV/TV in female 10-week-old mice is quite strong (especially in the Ctsk-cre mice). Did the authors detect extramedullary hematopoiesis? Are those mice iron deficient also in the liver? How is their systemic iron levels?

5. The lack of osteoclast function during Tfr1-deficiency would be beneficial in a more clinical context in e.g. ameliorating bone loss via ovariectomy. I think performing such an experiment (with the Tfr1;Ctsk-cre mice) would add more clinical benefit to the paper.

6. How do the authors explain that Tfr1 deficiency in osteoclast precursors has less effects than using the Ctsk-cre mice? I would assume that using the Lysm-cre, all osteoclasts – also the later stages – are affected, whereas in the Ctsk-cre, only the mature ones are. Thus, I am not sure why the Lysm-cre don´t also show a stronger effect.

7. Figure 4: to me, it actually looks like there are fewer osteoclasts in the cKO mice… the number of mice is not that high for histology, which tends to have high standard deviations. Did the authors also check the BFR in those mice? Was it also a bit increased, like in the Lysm-cre mice?

8. A WB should be shown validating the reduction of OXPHOS complex I-V expression. Similarly, it would be good to show a verification of other key proteins involved in cytoskeletal reorganization that were identified in the proteomic studies.

9. It is not clear to me how Mitotracker/Mitosox MFI and mitochondrial membrane potential were quantified? Why are there so many dots? How many cells and/or mitochondria were quantified? Was the MFI of each individual cell measured? Please add more details.

10. The link between Tfr1 and cytoskeleton organization is not quite clear. Is that connection e.g. iron-dependent and/or dependent on non-canonical functions of Tfr1? If iron-dependent, how does iron affect the regulation of WRC/Arp2/3 complex?

11. I personally think that Suppl. Figure 1 should be included into the main figures as it shows the first mechanistic part. I would rather suggest to potentially combine Figure 1+3 and/or 2+4 (or put parts of them into the supplement).

[Editors' note: further revisions were suggested prior to acceptance, as described below.]

Thank you for resubmitting your work entitled "Transferrin receptor 1-mediated iron uptake regulates bone mass in mice via osteoclast mitochondria and cytoskeleton" for further consideration by *eLife*. Your revised article has been evaluated by Mone Zaidi (Senior Editor) and a Reviewing Editor.

While the authors have added a substantive amount of work to significantly improve this manuscript, there are some remaining issues that need to be addressed, still a few places that need further explanation, a few pieces of data that do not evidently communicate what the authors claim:

1) While Ctsk mRNA expression is increased in Tfr1f/f Cre+ mice, protein concentration is decreased. This is unclear and presumptively, the focus is on the protein concentration. It is unclear what this means in total but the authors state that this indicates that Tfr1 deletion in osteoclasts has minimum effect on osteoclast differentiation. This needs to be clarified further in the results and Discussion sections.

2) Tfr1 loss appears to increase the number of osteoclasts with fewer nuclei and decreased fusion. The authors state that this impacts osteoclast function (decreased) rather than number or differentiation. This is fine as they show good evidence of decreased osteoclast functional endpoints. However, later in the manuscript, osteoblast data is also alluded to but not presented, noting only increased serum P1NP and Runx2 expression (Figure 3 – suppl Figure 2) and a comment suggesting that the phenotype in Tfr1f/f;Lyz2-Cre mice therefore maybe the consequence of altered osteoblast function. The response to Reviewer 1, point #8 is thus confusing and requires further clarification. Perhaps the authors are stating that the altered osteoclast function leads to an altered ability of the osteoclasts to suppress osteoblasts? This should be made clear in the discussion at least as speculation if the data cannot be shown fully. Response to Reviewer 2 point #4 is confusing with circular logic along the same lines.

3) Figure 6D needs a quantification. The authors state that Tfr1 loss results in decreased energy metabolism but there is no obvious evidence of this in osteoclasts in this figure (maybe BMM are bit more decreased in Tfr1-/- cells). Therefore, it is erroneous to state that is is restored by hemin as there is no evidence of a defect or visually appreciable differences relative to control in any of the wells. Please clarify.

4) While OVX Tfr1f/f;LysMCre mice exhibit a persistent difference from OVX control, the authors state that the phenotype observed in non-OVX mice is estrogen dependent (page 9, line 227). This is conceptually inaccurate in my understanding as if it were estrogen dependent, the difference would be abrogated after OVX which it is not. It in fact supports the opposite conclusion, that the phenotype is NOT estrogen dependent. Please clarify in discussion.

5) Figure 9C and 9D demonstrate the effect of Hem1 overexpression. The authors note "rescue" but this is only partial and a more appropriate conclusion would be that other mechanisms of iron delivery can reverse, at least partially, the effect of iron restriction induced by Tfr1 loss.

6) Unclear if there is a difference between 10uM hemin in control vs. Tfr1-/- cells in Figure 5A, 5C, and 5D. A preserved difference would suggest that there is incomplete rescue even at the highest hemin dose because osteoclasts may not have every option for iron import available in other cells. This is important to expand in the Discussion section.

7) Figure 9B requires quantification to corroborate the assertions made by the authors.

---

## [Author Response]

Essential revisions:The reviewers found significant merit in the work. However, the evaluation identified sufficient weaknesses that prevent publication of this work in the present form. The main shortcomings of the presented work include:1) The relatively underdeveloped in vitro data which does not convincingly support the conclusions.

To further support our conclusion that loss of Tfr1 in osteoclast lineage cells has little effects on differentiation but attenuates cytoskeleton organization and inhibits activation/function of mature osteoclasts, we performed the following extended in vitro experiments. The new data are included in new Figure 4, Figure 6, and Figure 9.

i. qPCR detection of mRNA expression of osteoclast marker genes, including *Acp5* (encoding TRAP), *Ctsk* (encoding Cathepsin K), *Nfatc1* (encoding osteoclast master transcription factor Nfatc1), and *Dcstamp* (encoding osteoclast fusion regulator DC-Stamp), in control and Tfr1^ΔlysM^ bone marrow monocytes, pre-osteoclasts, and mature osteoclasts. As shown in Figure 4B, the mRNA levels of *Acp5*, *NFATc1*, and *dcstamp* in Tfr1^ΔlysM^ osteoclast precursor and mature cells were similar to those of respective control cells. The mRNA expression of *Ctsk* in Tfr1^ΔlysM^ pre-osteoclasts and mature osteoclasts was slightly higher than that in control cells. These results indicate that deletion of Tfr1 in osteoclast myeloid precursors has minimum effects on osteoclast differentiation as indicated in Figure 4A.

ii. We repeated western blotting for Nfatc1 and Cathepsin K and quantified the specific bands of these osteoclast markers by densitometry using NIH Image J software. As demonstrated in the right panels of Figure 4C and Figure 4—figure supplement 1B, deletion of Tfr1 had no effects on the protein level of Nfatc1 in osteoclast precursor and mature cells, whereas the level of Cathepsin K in Tfr1^ΔlysM^ pre- and mature-osteoclasts was slightly decreased compared to control and Tfr1-f/f;c/+ osteoclasts.

iii. We counted the number of osteoclasts with 3-10, 10-30, and >30 nuclei, respectively. As shown in Figure 4D, the number of osteoclasts with 3-10 nuclei in Tfr1^ΔlysM^ culture was higher than control and there was no difference in the number of osteoclasts with 10-30 nuclei between control and Tfr1^ΔlysM^ cultures. Only the number of osteoclasts with > 30 nuclei in Tfr1^ΔlysM^ culture was lower than control although the mRNA expression of osteoclast fusion gene *Dcstamp* was similar in control and Tfr1^ΔlysM^ osteoclasts. The exact mechanism(s) of decreased fusion in the late stage of osteoclastogenesis in *Tfr1*^ΔlysM^ culture need further investigation in future.

iv. To consolidate the finding from quantitative proteomic results that cellular iron regulates the components of mitochondrial respiration chain, we did immuno-blotting to detect the changes of mitochondrial respiratory complexes in control and Tfr1^ΔlysM^ monocytes and mature osteoclasts treated with either vehicle or 10μM hemin using a cocktail of antibodies recognizing the representative components of mitochondrial respiratory complex I-V. As shown in Figure 6D, there was an obvious decrease in the mitochondrial complex II and III in Tfr1^ΔlysM^ monocytes and hemin stimulated the mitochondrial complex I and II in control and Tfr1^ΔlysM^ osteoclasts, indicating that Tfr1-mediated iron uptake plays an important role in mitochondrial oxidative phosphorylation pathway.

v. To identify the mechanisms by which Tfr1 regulates osteoclast WRC complex and cytoskeleton, we examined the activation (phosphorylation) of protein tyrosine kinase c-Src which plays a pivotal role in osteoclast cytoskeleton organization downstream of integrin and M-CSF as well as RANKL signaling pathways in control and Tfr1^ΔlysM^ osteoclasts. We also assayed the activation of small GTPases Rac1 and Arf1 that are upstream stimulators of WAVE-regulating complex (WRC) in responses to M-CSF and RANKL in control and Tfr1^ΔlysM^ osteoclasts using GST-pulldown kits from Thermo-Fisher Scientific Inc. As shown in Figure 9B, we found that the phosphorylation of c-Src and the GTP-bound (active form) of Rac1, but not GTP-bound Arf1, stimulated by M-CSF were declined in Tfr1^ΔlysM^ osteoclasts. In contrast, Tfr1-deficiency had little effects on RANKL-activated c-Src, Rac1, and Arf1 in osteoclasts.

vi. We repeated the Hem1 overexpression experiment with proper controls suggested by the reviewers. As demonstrated in Figure 9C and 9D, overexpression of HA-tagged Hem1 enhanced the number of activated osteoclasts in control culture and rescued the cytoskeletal defect in Tfr1^ΔlysM^ osteoclasts.

2) The concern that the mouse models result in decreased Ctsk expression which may along at least in part explain the phenotype;

As shown in Figure 4B and 4C, although the mRNA expression of *Ctsk* in Tfr1^ΔlysM^ pre-osteoclasts and mature osteoclasts detected by qPCR was slightly higher than that in control cells, the quantification of western blots demonstrated that the levels of Ctsk protein in Tfr1^ΔlysM^ pre- and mature-osteoclasts were slightly decreased compared to control cells. This decreased Ctsk expression might contribute attenuated bone resorption in Tfr1^ΔlysM^ osteoclasts. It should be pointed out that Ctsk heterozygous mice had no defects in osteoclast function (Gowen, et al., JBMR 1999). The phenotypes of osteoclasts in vitro and skeletal homeostasis in Tfr1^ΔlysM^ mice are mainly resulted from the decreased energy metabolism and defect in cytoskeleton organization in Tfr1^ΔlysM^ osteoclasts.

3) The potential of evaluation in an ovarectomized mice and / or mice at different ages to extend the translational potential and underlying mechanism of these findings, respectively.

In addition to 10 weeks old post-pubertal mice, we have examined the skeletal phenotypes of 3-weeks-old pre-pubertal and 6 months old sham and ovariectomized control and Tfr1-flox;LysM-Cre mice by μCT. These data are presented in Figure 2 and Figure 2—figure supplement 4-5. The results are incorporated in the third paragraph on page 8-9. The possible mechanisms are discussed in the second paragraphs on pages 20-21.

4) Some degree of evaluation of systemic iron metabolism and erythropoiesis.

We have measured the serum levels of total iron and iron-regulating hormone hepcidin. In addition, we have counted the number of red blood cells in peripheral blood from both male and female control and Tfr1-flox;LysM-Cre mice. These data are presented in the Figure 2—figure supplement 3. The results have been added in the second paragraph on page 8, which demonstrate that loss of Tfr1 in myeloid lineage cells has no effects on systemic iron homeostasis and erythropoiesis.

5) Specifically, the in vitro data can be extended to more fully explore osteoclastogenesis and in vivo data by experimentally addressing the observed gender disparity and specificity of effect in the trabecular region of long bones.

See specific point-by point responses below to individual reviewer’s critiques related to this issue.

Reviewer #1 (Recommendations for the authors):Some additional questions and clarifications are delineated below.1. It is unclear why only or mostly female osteoclast-selective TFR1-null mice exhibit a more robust phenotype, with increased bone mass in long bones. The authors discuss this point in the Discussion section but the proposed experiment, to evaluate whether hepcidin expression is altered in the liver or hepcidin concentration changes in the serum and IHC using anti-FPN1 antibody in osteoclasts or at least cellular iron concentration would be useful to corroborate the hypothesis of estrogen-mediated changes in the hepcidin:ferroportin axis. Do osteoclasts even express FPN1 to make such a hypothesis viable? There is some data on testosterone and hepcidin regulation (Latour Hepatology 2014) that may be worth mentioning or discussing in the setting of the authors hypothesis about differential gender-related results. Finally, are the differences between male and female mice significant in Figure 2? Are these differences in the expected range for male and female mice?

The serum levels of iron and hepcidin in 10-week-old male and female control and *Tfrc*-flox; *Lyz2*-Cre mice are similar (see Figure 2—figure supplement 3). It is well known that the trabecular bone mass of male mice is higher than their age-matched female mice in the adults. Since Tfr1-deficiency results in significant increase in trabecular bone mass solely in female but not male mice, there is no difference of bone mass between female *Tfrc*-flox;*Lyz2*-Cre mice and male *Tfrc*-flox;*Lyz2*-Cre as well as male control mice. This phenomenon has been added on page 9.

2. It is similarly unclear why only long bones but not vertebrae reveal these differences. Can the authors speculate and discuss this point in the Discussion section?

Mouse genetic studies have clearly established that the genetic regulation of peak bone mass development is skeletal site dependent. Thus, it is not surprising that Tfr1 deficiency exerts different effects on the skeletal phenotype of long bones versus vertebra. These differences are discussed in the context of what is known in the literature on osteoclast heterogeneity and genetic regulation of bone development at these two sites on pages 22-23. Our qPCR result demonstrated that deletion efficiency of Tfr1 gene, *Tfrc*, in vertebra is less potent than femurs (Figure 2—figure supplement 9).

3. The authors suggest that they have effectively excluded an effect from TFR2 on osteoclast function. However, this is inaccurate and needs to be communicated differently. At best, the authors have confirmed what has long been understood that TFR2 serves as an iron sensor rather than importer of iron. This should be restated in all mention of TFR2 in the manuscript. Furthermore, Rauner et al. published an important manuscript looking at the role of TFR2 in bone, demonstrating that TFR2 competes with BMPR for binding BMP2 and TFR2:BMP binding is enhanced in the presence of Fe-TF, together suggesting that presence of iron may favor bone resorption in the presence of TFR2. Something akin to this would benefit the readership and should be included in the manuscript as it pertains to the data on TFR1 here.

We agree. Although the results in Figure 1B indicate that Tfr1-mediated iron uptake is a major route for iron acquisition in osteoclasts, we did not have direct evidence regarding the effects of Tfr2 on osteoclast iron uptake and function. The statement of “The role of Tfr2 in osteoclast iron uptake and function needs to be further elucidated in the future” has been added in the second paragraph on page 7.

4. Figure 1, Y axis on mRNA expression panels is unclear what the control is. Typically, expectation is of change from baseline (1) in control. Minor: panel for Flvcr2 is missing from legend.

The information has been added in Figure 1A and its legend.

5. Figure 1 B, what mechanism leads to some cellular iron in TFR1 fl/LysM Cre OC cells? This may be the reason that the results on differentiation are not seen in the current work relative to Ishii et al. 2009. Please explain and include in the Discussion.

The residual cellular iron in Tfr1-deficient osteoclasts may be obtained through uptake of heme because the heme transporter Hcp1 is up-regulated during osteoclast differentiation and the fact that hemin but not ferric ammonium citrate (FAC, which is absorbed through NTBI) is able to rescue the Tfr1-deficient osteoclast phenotypes (Figure 5). While Ishii et al. reported that knockdown of Tfr1 expression in bone marrow monocytes by short-hairpin RNA (shRNA) inhibits osteoclast differentiation and transferrin promotes osteoclastogenesis in vitro, the treatment of mice with iron chelator has no effects on osteoclast number in vivo. Iron chelator inhibits iron effect totally and not specifically to Tfr1. The use of mice with conditional deletion of Tfr1 gene specifically in osteoclasts to investigate the role of Tfr1 mediated iron transport function in osteoclasts has not been done previously. The information has been added on page 23.

6. Figure 3 A-E is redundant and can be moved to the Supplement. In general, the data on LysM-cre mice is distracting and does not provide substantive contributing information to the main point. Consider moving all LysM-cre data to the supplement to better focus the manuscript.

Please see our response to General Comment #1, both the *Lyz2*-Cre and *Ctsk*-Cre models have advantages and disadvantages. Since *Lyz2*-Cre is relatively more specific than *Ctsk*-Cre in gene-deletion in osteoclast lineage cells (see the reference in Yang W et al., *Nature*, 2013), we moved *Tfrc*-flox;*Ctsk*-Cre results to the Figure 2—figure supplement 6-7 and Figure 3—figure supplement 1/3.

7. Figure 3G, the difference in female mice is driven by 3 mice with elevated values; are these 3 mice in any way different from the others? Are they outliers?

Since the other histological parameters of these three mice (See the Figure 3 source data file) are not quite different from other mice, we think that the data in Figure 3G reflects the high variation in in vivo data. We decide to keep the data of these three mice.

8. Figure 3 states that there is no impact on osteoclast number and bone resorption in mice but while BFR is unchanged in Figure 3J, serum P1NP is increased. How do the authors reconcile this confusing result?

The calculation of BFR in Osteomeasure software is normalized to bone surface (BS). Since there is a significant increase in trabecular bone mass (also in BS) in *Tfrc*-flox;*Lyz2*-Cre female mice, the total level of P1NP (procollagen type 1 N-terminal propeptide, a specific marker of type 1 collagen deposition during bone formation) is up in these mice. This is supported by increased mRNA level of osteoblast transcription factor Runx2 in *Tfrc*-flox;*Lyz2*-Cre female mice (Figure 3—figure supplement 2). See also the reply in #14 of Reviewer 3.

9. Given the number of figures, it would be worthwhile to consolidate and reorganize to move some additional elements into the supplementary figures. For example, there is no reason in the mind of the reviewer to include both male and female samples in vitro. Especially since there are no visible differences, please move one gender to the supplement.

We moved all the male in vitro and in vivo results to the Supplemental Figures. The male osteoclasts have similar phenotypes to female cells in vitro. Thus, the differences of in vivo phenotypes between male and females are resulted from the distinct effects of sex hormones on osteoclasts in vivo.

10. Figure 6 title suggests that there are no differences in markers of differentiation, e.g. Nfatc1 and Ctsk, but both of these appear to be decreased in f/f;c/c samples, both female but especially male (especially since tubulin appears greater in those samples). Please provide the quantification here.

During the revision of this manuscript, we examined the mRNA expression of osteoclast marker genes by qPCR in control and Tfr1^ΔlysM^ osteoclast lineage cells. As shown in Figure 4B, the mRNA levels of *Acp5*, *Nfatc1*, and *dcstamp* in Tfr1^ΔlysM^ osteoclast precursor and mature cells were similar to those of respective control cells. The mRNA expression of *Ctsk* in Tfr1^ΔlysM^ pre-osteoclasts and mature osteoclasts was even higher than that in control cells. We also quantified the western blots of Nfatc1 and Ctsk. Deletion of Tfr1 in both male and female mice had no effects on the protein level of Nfatc1 in osteoclast precursor and mature cells, whereas the level of Ctsk in Tfr*1*^ΔlysM^ pre- and mature-osteoclasts was slightly decreased compared to control and Tfr1-f/f;c/+ osteoclasts (the right panels in Figure 4C and Figure 4—figure supplement 1B).

11. Figure 12B and 12C do not include control cells with and without Hem1 overexpression. This is important since the mechanism may be completely different and yet additive to yield the observed results. Please edit.

Please see our response to General Comment #4.

12. The Introduction is too long and unfocused. Please edit.

The Introduction has been shortened.

13. The first half of the Discussion is a restating of the results and needs to be re-written to further extend the interpretation of the results without restating them.

We have revised the discussion to reduce redundancy with the Results section.

Reviewer #2 (Recommendations for the authors):This is a well-written study that examines the role of transferrin receptor 1 in osteoclast function and bone remodelling. The importance of iron homeostasis in bone turnover has gained momentum in recent years, most notably the role of transferrin receptor 2 in osteoblasts (Rauner et al., Nat Med 2019), however the physiological significance of iron and the mechanism(s) responsible for its uptake in osteoclasts remains poorly understood. As such, the present study detailing the physiological role of Tfr1 in osteoclasts would likely be of considerable interest to the field, complementing existing studies in osteoblasts, but also reconciling the physiological contribution of cellular iron handling on the other side of the bone remodelling equation. The strengths of this study lie the multiple osteoclast-specific mouse lines generated to interrogate the role of Tfr1 in both osteoclast-lineage cells and mature osteoclasts along with the suite of complementary cell biological, biochemical and proteomic analyses used to dissect its contribution in osteoclasts. Nonetheless, there remain some weaknesses, particularly concerning the expressivity of the bone phenotype and mechanistic link between Tfr1 and the cytoskeleton that should be considered to strengthen the overall resolve of the conclusions drawn.I have the following comments:1. The studies conducted on LysoM-Cre mice appear to be well-executed and carefully controlled. The skeletal phenotype is however complex, being restricted to trabeculae of the long bones of female mice, but not the axial skeleton (e.g. vertebrae) at 10-weeks of age. The authors rationalise that the unexpected sexual dimorphism is associated with an intersection between estrogen and iron export, which is reasonable, but they do not offer experimental evidence to support this. Further, the skeletal phenotyping is limited to a single time-point i.e. 10-weeks of age. As such, it would be beneficial if the authors could provide data in older animals to show whether the bone phenotype becomes more pronounced or resolves with age. It would also be useful for the reader if the authors could offer some explanation for the observed site-specific differences in the Discussion e.g. potential differences in osteoclast heterogeneity etc.

See the reply to #3 in Essential Revisions and the results in new Figure 2 and supplemental Figures 5-6. The explanations for the different phenotypes in trabecular verse cortical bone compartments and in long bone versus vertebra are discussed in the third paragraph on page 22.

2. Although widely employed as a Cre-transgenic mouse line to conditionally delete target genes in mature osteoclasts there are some caveats that should be considered to avoid over-interpretation of data arising from the cathespin K (Ctsk)-Cre lines. Foremost, this cre-line is known to result in the loss one allele of Ctsk owing to partial replacement the Ctsk locus with Cre in the targeting cassette (Dallas et a., Genetics 2018). Thus, endogenous Ctsk expression is reduced. Considering that deficiency and mutations in Ctsk lead to high bone mass phenotype (Saftig et al., 1997) together with findings that Cstk K-Cre can also lead to unexpected germline deletion of genes (e.g. Winkeler et al., PLos 2012), any data arising from these mice need to be interpreted with a degree of caution. The authors have rightly attempted to control for this using littermates mice on heterozygous KI (Ctsk-Cre/+), which is appropriate although these mice are on a mice mixed background C57BL6/129 which can complicate skeletal phenotypes. Overall, these underlying complexities may account, in part, for the discrepancies observed between the LysoM and Ctsk-cre males and females, with Ctsk-Cre leading to a more pronounced phenotype. As a minimum, these limitations need to be acknowledged and discussed. A minor oversight- for Figure 4B is that the mice sexes are unlabelled in the reconstructed uCT scans.

The advantages and disadvantages of *Lyz2*(*LysM*)-Cre and *Ctsk*-Cre models and the possible explanations for the different phenotypes between the *Lyz2*-Cre and *Ctsk*-Cre males and females have been added in and discussed in the second paragraph on page 22. See also the reply to the #1 of General Comments of Reviewer 1.

3. The in vitro data concerning the potential effects of Tfr1-deficiency on osteoclast different and mitochondrial metabolism are extensive and overall support the conclusion drawn. On the other hand the mechanism linking Tfr1 regulation of the osteoclast cytoskeleton remains largely descriptive and thus is comparatively underdeveloped. While the quantitative proteomics data goes some way to lend support to the notion that loss of Tfr1 leads to a global dysregulation of downstream Tfr1-associated protein pathways including those involved with cytoskeletal signalling/organization, the extent and diversity of proteins affected in osteoclasts >2000 proteins is remarkable. Can the authors please clarify how they arrived at the 1.2-fold threshold for these data. Does the 1.2-fold represent an arbitrary cut-off or does it reflect >1.2 Log2Fold, p<0.05? To validate these proteomic data the authors hone in on hem1 (a component of the Wave regulatory complex, WRC) and show that it partly restores the cytoskeletal spreading defect in Tfr1-deficient osteoclasts. This finding is interesting but would be bolstered by confirmation of a disruption of additional components of the WRC or downstream components such as Arf2/3 which has an established role in OC cytoskeletal organization in osteoclasts (Hurst et al., JBMR 2009). Considering the well-established roles of B3-integrin, c-SRC and Pyk2 in osteoclasts, it would be valuable to corroborate/or discount their direct involvement in these spreading and cytoskeletal defects by independently assessing their expression levels immunoanalytically in Tfr1-deficient osteoclasts.

The advantages and disadvantages of *Lyz2*(*LysM*)-Cre and *Ctsk*-Cre models and the possible explanations for the different phenotypes between the *Lyz2*-Cre and *Ctsk*-Cre males and females have been added in and discussed in the second paragraph on page 22. See also the reply to the #1 of General Comments of Reviewer 1.

Reviewer #3 (Recommendations for the authors):1. The authors state that Tfr1 does not inhibit osteoclast differentiation, but just spreading and resorption. I do not agree with this statement. By "spreading", do the authors mean "fusion"? I believe in any case that this is part of the differentiation process to a fully mature osteoclast. Also, even though it is stated that osteoclasts have been counted as TRAP-positive and with more than 3 nuclei, I know from experience that it is tricky to identify the nuclei just using TRAP staining, especially in such small cells as presented in Suppl. Figure 9A. I believe it may be more accurate to assess the number of osteoclasts in this case with so many small ones using IF and DAPI staining for the nuclei. Using that, the authors could also count how many osteoclasts have e.g. 3-10 or 10-50 nuclei (giving insights into the fusion process). Finally, I believe more markers for osteoclasts could be used to address this point of fusion/spreading, such as DC-STAMP, SWAP70, or CTSK. In Suppl. Figure 9, e.g., it looks like CtsK expression is reduced in the Tfr1;Lysm-cre mice, although the WB is a bit over-exposed, so it is hard to tell… quantification would be helpful.

The word “spreading” is often used to describe the cellular process in which the morphology of multinucleated osteoclasts changes from the irregular-shaped status characterized by scattered distribution of podosomes to a round-shaped status featured by the peripherally localized podosome-belt when osteoclasts are cultured on glass coverslips or plastic culture dishes. Thus, the spreading process in osteoclasts is regulated by cytoskeleton organization. The defects in osteoclast spreading are associated with decreased osteoclast activation and bone resorption. The reviewer’s point is correct that the “fusion” is a part of osteoclast differentiation process where mononuclear osteoclast precursor cells are fused to form multinucleated osteoclasts. This process is regulated by osteoclast fusion genes including DC-Stamp induced by RANKL and Nfatc1.

In response to the reviewer suggestion, we have now quantitatively measured the mRNA expression of osteoclast marker genes in control and Tfr1^ΔlysM^ osteoclast precursor and mature cells. As shown in Figure 4B, the mRNA levels of *Acp5*, *Nfatc1*, and *Dcstamp* in *Tfr1*^ΔlysM^ osteoclast precursor and mature cells were similar to those of respective control cells. The mRNA expression of *Ctsk* in Tfr1^ΔlysM^ pre-osteoclasts and mature osteoclasts was slightly higher than that in control cells. These results indicate that deletion of Tfr1 in osteoclast myeloid precursors has minimum effects on osteoclast differentiation as we observed in Figure 4A (see also the reply in #*i* of Essential Revisions #1).

Per the reviewer’s suggestion, we counted the number of osteoclasts with 3-10, 10-30, and >30 nuclei, respectively. As shown in Figure 4D, the number of osteoclasts with 3-10 nuclei in Tfr1^ΔlysM^ culture was higher than control and there was no difference in the number of osteoclasts with 10-30 nuclei between control and Tfr1^ΔlysM^ cultures. Only the number of osteoclasts with > 30 nuclei in Tfr1^ΔlysM^ culture was lower than control although the mRNA expression of osteoclast fusion gene *Dcstamp* was similar in control and Tfr1^ΔlysM^ osteoclasts. The exact mechanism(s) of decreased fusion in the late stage of osteoclastogenesis in Tfr1^ΔlysM^ culture need further investigation in future.

The suppl Figure 9 was in our previous but not in this submission. Instead, we repeated western blotting for Nfatc1 and Ctsk and quantified the specific bands of these osteoclast markers by densitometry using NIH Image J software. As demonstrated in the right panels of Figure 4C and Figure 4—figure supplement 1B, deletion of Tfr1 had no effects on the protein level of Nfatc1 in male and female osteoclast precursor and mature cells, whereas the level of Ctsk in Tfr1^ΔlysM^ pre- and mature-osteoclasts was slightly decreased compared to control and Tfr1-f/f;c/+ osteoclasts in both gender. This slight decrease in Ctsk protein level in Tfr1^ΔlysM^ osteoclasts might be caused by the fewer number of large osteoclasts with > 30 nuclei in Tfr1^ΔlysM^ culture than in control.

2. It appears that the appendicular skeleton is more affected than the axial skeleton. Did the authors check the expression of Tfr1 in e.g. femoral vs. vertebral bone using immunohistochemistry? Is there a difference in expression pattern that may explain the altered susceptibility of the sites to Tfr1 deletion? Similarly, it would be interesting to check Tfr1 in trabecular vs. cortical bone – as those also seem to be differentially affected.

As discussed on page 22, the exact explanation(s) for distinct effects of Tfr1-deficiency on the appendicular *vs* axial bones and the trabecular *vs* cortical bones are not available. Unfortunately, the anti-Tfr1 antibody we used in Figure 4A does not work for immunohistochemistry. By qPCR, we demonstrated that the deletion efficiency of Tfr1 gene, *Tfrc*, in vertebra is less than femurs (Figure 2—figure supplement 9).

3. Along those lines, is the expression of Tfr1 in osteoclasts (bone) lower in young animals compared to adult ones? Could that explain the more severe effects of iron deficiency in adult mice compared to the 3-week-old ones?

As mentioned above, we could not answer this question because the anti-Tfr1 antibody does not work for immunohistochemistry on EDTA-decalcified, paraffin-embedded bone tissue sections.

4. The increase in BV/TV in female 10-week-old mice is quite strong (especially in the Ctsk-cre mice). Did the authors detect extramedullary hematopoiesis? Are those mice iron deficient also in the liver? How is their systemic iron levels?

See the reply for #4 in Essential Revisions. Loss of Tfr1 in osteoclasts has no effects on systemic iron homeostasis and erythropoiesis as detected by serum iron and peripheral blood RBC count. There is no splenomegaly in Tfr1^ΔLysM^ and Tfr1^ΔCtsk^ mice (data not shown). Serum hepcidin, a liver-derived iron-regulating hormone, levels are similar between control and Tfr1-deficient mice, thus suggesting that there is no iron deficiency in liver. However, we have not measured liver iron level in mice with disruption of Tfr1 in osteoclasts.

5. The lack of osteoclast function during Tfr1-deficiency would be beneficial in a more clinical context in e.g. ameliorating bone loss via ovariectomy. I think performing such an experiment (with the Tfr1;Ctsk-cre mice) would add more clinical benefit to the paper.

When the lab moved to California in 2018, we discontinued the *Tfrc*-flox;*Ctsk*-Cre strain. We performed OVX on 4.5-monthold control and Tfr1^ΔLysM^ female mice. After 6-week post OVX, the mice were sacrificed, and trabecular bone mass and cortical bone thickness were measured by μCT. As shown in Figure 2, estrogen deficiency resulted in decreased trabecular bone in both female control and Tfr1^ΔlysM^ mice, indicating that Tfr1 in osteoclasts does not influence bone loss in OVX mice. This finding is discussed on page 20-21.

6. How do the authors explain that Tfr1 deficiency in osteoclast precursors has less effects than using the Ctsk-cre mice? I would assume that using the Lysm-cre, all osteoclasts – also the later stages – are affected, whereas in the Ctsk-cre, only the mature ones are. Thus, I am not sure why the Lysm-cre don´t also show a stronger effect.

The distinct effects of Tfr1-deficiency by *Lyz2*-Cre and *Ctsk*-Cre may be due to the different in vivo efficiency of these two Cre lines and may also be caused by the off-target effects of the Cre drivers as mentioned in the reply for the #1 critic in General Comments of Reviewer 1.

7. Figure 4: to me, it actually looks like there are fewer osteoclasts in the cKO mice… the number of mice is not that high for histology, which tends to have high standard deviations. Did the authors also check the BFR in those mice? Was it also a bit increased, like in the Lysm-cre mice?

The number of osteoclasts in control and Tfr1^ΔCtsk^ mice is the same. The staining in cKO mice was faint by unknown reason. We only did histology in decalcified, paraffin-embedded bone sections for TRAP staining. We did not do tetracycline-labeling in these mice.

8. A WB should be shown validating the reduction of OXPHOS complex I-V expression. Similarly, it would be good to show a verification of other key proteins involved in cytoskeletal reorganization that were identified in the proteomic studies.

See Figure 6B and the reply in #*iv* of Essential Revisions above. We had difficulties in finding antibodies with high specificity and affinity against the components of WRC complex by western blotting. Instead, we found that the M-CSF stimulated activation of c-Src and Rac1, which are upstream regulators of WRC complex, were decreased in Tfr1^ΔlysM^ osteoclasts (Figure 9B). See the reply in #*v* of Essential Revisions above and in #10 below.

9. It is not clear to me how Mitotracker/Mitosox MFI and mitochondrial membrane potential were quantified? Why are there so many dots? How many cells and/or mitochondria were quantified? Was the MFI of each individual cell measured? Please add more details.

Please see “Mitochondrial mass, ROS production, and membrane potential measurements” in the Materials and methods section for detailed methodology. The following information has been added (See grey highlights on page 32).

The epifluorescent images of 10-15 randomly chosen areas per group (control or knockout) were taken in a blinded fashion. Each area had between 7-15 cells. The images were analyzed with Image J software also in a blinded fashion. The fluorescent intensities of cells were individually measured, and the results were presented as mean fluorescent intensity (MFI) per cell in arbitrary units.

10. The link between Tfr1 and cytoskeleton organization is not quite clear. Is that connection e.g. iron-dependent and/or dependent on non-canonical functions of Tfr1? If iron-dependent, how does iron affect the regulation of WRC/Arp2/3 complex?

As shown in Figure 5C and 5D, 10μM hemin (ferric chloride heme), which provides iron through the alternative heme uptake in the absence of Tfr1, rescued the cytoskeleton organization defects (podosome-belt formation in osteoclasts cultured on glass coverslips and the actin-ring formation in osteoclasts cultured on bovine cortical bone slices, see also Figure 4D and 4E) in Tfr1^ΔlysM^ osteoclasts. These results indicate that Tfr1-mediated cellular iron homeostasis regulates osteoclast cytoskeleton organization although we don’t have evidence whether the non-canonical functions of Tfr1 participates this process at the moment. To identify the mechanisms by which Tfr1-mediated iron uptake regulates WRC/Arp2/3 complex, we turned to its upstream activators, the small GTPases Rac1/Arf1 and the protein tyrosine kinase c-Src which plays a central role in activating Rac1 and osteoclast cytoskeleton stimulated by the integrins, M-CSF as well as RANKL. As shown in Figure 9B, we found that the phosphorylation (activation) of c-Src and the GTP-bound (active form) of Rac1 but not GTP-bound Arf1 stimulated by M-CSF were declined in Tfr1^ΔlysM^ osteoclasts. In contrast, Tfr1-deficiency had little effects on RANKL-activated c-Src, Rac1, and Arf1 in osteoclasts.

11. I personally think that Suppl. Figure 1 should be included into the main figures as it shows the first mechanistic part. I would rather suggest to potentially combine Figure 1+3 and/or 2+4 (or put parts of them into the supplement).

The Supplemental Figure 1 has been changed to Figure 1. The previous Figure 1 and Figure 2 are now Figure 2 and Figure 3, respectively. We moved previous Figure 3 and Figure 4 into the supplemental Figures. They are now Figure 2—figure supplement 6 and Figure 3—figure supplement 3, respectively.

[Editors' note: further revisions were suggested prior to acceptance, as described below.]

While the authors have added a substantive amount of work to significantly improve this manuscript, there are some remaining issues that need to be addressed, still a few places that need further explanation, a few pieces of data that do not evidently communicate what the authors claim:1) While Ctsk mRNA expression is increased in Tfr1f/f Cre+ mice, protein concentration is decreased. This is unclear and presumptively, the focus is on the protein concentration. It is unclear what this means in total but the authors state that this indicates that Tfr1 deletion in osteoclasts has minimum effect on osteoclast differentiation. This needs to be clarified further in the results and Discussion sections.

We apologize for lack of clarity on the Ctsk mRNA expression and protein data. Cathepsin k is a well-accepted marker of osteoclast activity. Thus, our data on the reduced CTSK protein levels in Tfr1^ΔLysM^ mice suggest reduced osteoclast activity. The result and conclusion have been revised on page 12 and page 13.

2) Tfr1 loss appears to increase the number of osteoclasts with fewer nuclei and decreased fusion. The authors state that this impacts osteoclast function (decreased) rather than number or differentiation. This is fine as they show good evidence of decreased osteoclast functional endpoints. However, later in the manuscript, osteoblast data is also alluded to but not presented, noting only increased serum P1NP and Runx2 expression (Figure 3 – suppl Figure 2) and a comment suggesting that the phenotype in Tfr1f/f;Lyz2-Cre mice therefore maybe the consequence of altered osteoblast function. The response to Reviewer 1, point #8 is thus confusing and requires further clarification. Perhaps the authors are stating that the altered osteoclast function leads to an altered ability of the osteoclasts to suppress osteoblasts? This should be made clear in the discussion at least as speculation if the data cannot be shown fully. Response to Reviewer 2 point #4 is confusing with circular logic along the same lines.

We apologize for not clearly stating the osteoclast-osteoblast connection to the observed skeletal phenotype. Loss of Tfr1 expression in osteoclasts lead to decreased osteoclast function as shown by reduced CTSK protein levels and osteoclast functional data. Based on what is known about osteoclast-mediated coupling of bone formation, one would predict that the reduced osteoclast activity in Tfr1f/f Cre+ mice should reduce bone formation. By contrast, the increased serum PINP levels and Runx2 expression in these mice suggest increased bone formation. Thus, we don’t have an explanation at present if the increased bone formation is a direct consequence of inhibition of osteoclast activity or by some other mechanisms. Further studies are needed to address this issue.

3) Figure 6D needs a quantification. The authors state that Tfr1 loss results in decreased energy metabolism but there is no obvious evidence of this in osteoclasts in this figure (maybe BMM are bit more decreased in Tfr1-/- cells). Therefore, it is erroneous to state that is is restored by hemin as there is no evidence of a defect or visually appreciable differences relative to control in any of the wells. Please clarify.

Quantitative data of Figure 6D is added as Figure 6—figure supplement 1. The results are revised based on this quantification. Although Figure 6D provides complementary evidence for proteomic data (shown in Figure 6B and 6C) by simultaneously demonstrating the changes of mitochondrial respiratory complexes in vehicle and hemin treated control and Tfr1^ΔLysM^ bone marrow monocytes and osteoclasts, it should point out that the antibody cocktail used in this immunoblotting contains antibodies against only one component of each of five mitochondrial respiratory complexes as indicated in Figure 6D. Thus, the result of Figure 6D could not reflect complete changes of mitochondria in Tfr1-deficient osteoclast lineage cells. This limitation of the assay has been added on page 16. We agree that Figure 6 does not provide strong evidence to support the conclusion that loss of Tfr1 in osteoclasts attenuates energy metabolism. This conclusion is removed to Figure 8.

4) While OVX Tfr1f/f;LysMCre mice exhibit a persistent difference from OVX control, the authors state that the phenotype observed in non-OVX mice is estrogen dependent (page 9, line 227). This is conceptually inaccurate in my understanding as if it were estrogen dependent, the difference would be abrogated after OVX which it is not. It in fact supports the opposite conclusion, that the phenotype is NOT estrogen dependent. Please clarify in discussion.

We agree that the OVX data in Figure 2 could not fully support the conclusion that the high trabecular bone mass phenotype in Tfr1^ΔLysM^ mice is estrogen dependent though showing obvious gender-dependent difference. We revised the results and conclusion in the contest of what we know about estrogen regulation of bone modeling and remodeling during development and in adulthood in the Results and Discussion sections on page 9 and page 23, respectively. It is now well established that estrogen-mediated mechanisms play a key role in mediating bone accretion during pubertal growth period. During this developmental period, estrogen is known to exert an anabolic effect on osteoblast besides its well-established role in inhibiting osteoclast functions. However, the loss of estrogen as seen during menopause or due to ovariectomy in adults lead to bone loss primarily due to increased bone resorption. As shown in Figure 2, trabecular bone volume of Tfr1^ΔLysM^ mice was increased at the post-pubertal age of 10 weeks but was not different at the prepubertal age of 3 weeks, thus suggesting that the skeletal phenotype gets manifested during pubertal and post-pubertal growth periods. To determine if loss of Tfr1 in osteoclasts affects OVX-induced bone loss, control and Tfr1^ΔLysM^ mice at 4.5-month of age were ovariectomized to mimic human postmenopausal osteoporosis and skeletal phenotype was determined at 6 months of age. While OVX induced bone loss in Tfr1^ΔLysM^ mice, the trabecular bone mass of Tfr1^ΔLysM^ mice was significantly higher than control mice after OVX. While these data suggest that loss of Tfr1 does not affect estrogen deficiency-induced bone loss, the issue of whether the skeletal phenotype in 10-week-old Tfr1^ΔLysM^ mice is dependent on pubertal increase in estrogen remains to be evaluated by performing OVX at prepubertal age of 3 weeks in future.

5) Figure 9C and 9D demonstrate the effect of Hem1 overexpression. The authors note "rescue" but this is only partial and a more appropriate conclusion would be that other mechanisms of iron delivery can reverse, at least partially, the effect of iron restriction induced by Tfr1 loss.

We agree. We have revised the conclusion to indicate that the rescue is partial and suggest that the other component(s) of the WRC complex or distinct pathways might also be involved in the Tfr1-mediated regulation of osteoclast cytoskeleton in the first paragraph on page 20.

6) Unclear if there is a difference between 10uM hemin in control vs. Tfr1-/- cells in Figure 5A, 5C, and 5D. A preserved difference would suggest that there is incomplete rescue even at the highest hemin dose because osteoclasts may not have every option for iron import available in other cells. This is important to expand in the Discussion section.

We agree. We have added the statistical analysis results of 10uM hemin-treated control vs. Tfr1^ΔLysM^ cultures in Figure 5A, 5C, and 5D. We have revised the conclusion to indicate that the rescue is partial and there are limited alternative iron acquisition pathways in osteoclasts as in other cells in the first paragraph on page 15.

7) Figure 9B requires quantification to corroborate the assertions made by the authors.

Quantitative data is added as Figure 9—figure supplement 1, and the results and conclusion are revised based on the quantitation in the first and second paragraphs on page 19.